# Identifiable Nonlinear Differentiable Causal Discovery via Independence and Adaptive Group Sparsity

Ruicong Yao [1] [2]   Tim Verdonck [2] [3]   Mihaela van der Schaar [4]   Jakob Raymaekers [3]

## Abstract

Differentiable approaches to causal discovery have shown promise in learning DAG structures via continuous optimization, but their theoretical guarantees are largely restricted to models with homoscedastic noise or known noise distribution. In particular, existing methods based on mean squared error fail to identify the true DAG when noise distributions are non-Gaussian and vary in scale. In this paper, we address this gap in nonlinear additive noise models (ANMs) with arbitrary noise. Our approach extends NOTIME (Berrevoets et al., 2025) which minimizes an independence criterion among the residuals. We show that the global minimizer of the independence criterion corresponds to the true underlying DAG up to additional constant edges in general ANMs. To recover the exact structure, we introduce an adaptive group lasso penalty that regularizes entire columns of the first-layer weight matrix of an MLP, enabling the selective pruning of constant edges in a functionally meaningful way. Empirically, our method achieves effective and stable performance across diverse noise types and variances, outperforming prior methods that lack identifiability guarantees in this setting.

## 1. Introduction

Understanding causal relationships from observational data is fundamental to scientific discovery, policy-making, and many real-world decision systems. Whether in epidemiology, economics, or machine learning applications, reasoning about interventions requires more than mere associations: it requires knowledge of the underlying causal mechanisms. These mechanisms are often modeled by Directed Acyclic Graphs (DAGs) (Pearl, 1995), where nodes represent variables and edges represent direct causal effects. Causal discovery seeks to recover such a DAG from data under certain assumptions, e.g. functional constraints (Peters et al., 2017).

A flexible and widely studied class of models that enable identifiability from purely observational data is the Additive Noise Model (ANM) (Peters et al., 2014). In an ANM, each variable is modeled as a function of its parents in the DAG, plus an independent noise term. For nonlinear ANMs, their structures become identifiable under mild conditions (Peters et al., 2014). This makes ANMs particularly appealing for causal discovery in practical settings where interventions are unavailable or costly. However, learning the correct DAG under ANM assumptions presents unique statistical and computational challenges, especially in the nonlinear, high-dimensional, and heteroscedastic regimes often encountered in real-world data.

Non-differentiable methods for learning ANMs, such as RESIT (Peters et al., 2014), CAM (Bühlmann et al., 2014), and SCORE (Rolland et al., 2022), rely on combinatorial search procedures to construct a causal ordering and then prune spurious edges by independence testing or sparsity regularization. While showing empirical robustness in performance, CAM and SCORE have additional requirements in their theoretical guarantees. In particular, CAM requires the functions $f_j$ to be additive in the parents $\mathbf{PA}_j$, and SCORE relies on Gaussianity. RESIT, on the other hand, adapts to general ANMs but scales poorly with dimensionality.

Recently, differentiable DAG learning has emerged as a powerful alternative, offering scalable and gradient-based optimization frameworks (Zheng et al., 2018; Lachapelle et al., 2020; Ng et al., 2020; Zheng et al., 2020; Bello et al., 2022; He et al., 2021; Chen et al., 2023; Nazaret et al., 2024; Berrevoets et al., 2025). These methods relax acyclicity constraints into smooth surrogates, enabling joint learning of both structure and parameters via continuous optimization. However, most existing differentiable methods adopt either mean squared error (MSE) or log-likelihood as the loss function, which limits their identifiability guarantees to narrow settings, typically models with

---

[1]Department of Applied Mathematics, Computer Science, and Statistics, Ghent University, Belgium [2]Department of Mathematics, KU Leuven, Belgium [3]Department of Mathematics, University of Antwerp, Belgium [4]DAMTP, University of Cambridge, UK. Correspondence to: Ruicong Yao <ruicong.yao@ugent.be>.

*Proceedings of the 43rd International Conference on Machine Learning*, Seoul, South Korea. PMLR 306, 2026. Copyright 2026 by the author(s).

Gaussian or equal-variance noise (Loh & Bühlmann, 2014; Kaiser & Sipos, 2022; Reisach et al., 2021; Ng et al., 2024; Berrevoets et al., 2025). For nonlinear ANMs with heterogeneous noise regimes, these methods may fail to recover the true causal structure, as the loss function no longer aligns with the independence assumptions of the generative model. While Berrevoets et al. (2025) showed that any loss function based on a consistent joint dependence measure is valid for identifying the true DAG under Linear Non-Gaussian Acyclic Model (LiNGAM), its validity under general nonlinear causal models is unknown. Meanwhile, initialization is another challenge for differentiable methods to achieve good performance (Ng et al., 2024; Berrevoets et al., 2025), since the convexity of the loss functions is usually not guaranteed. Nevertheless, it remains not well-explored in the context of nonlinear causal discovery.

In this work, we extend the approach of Berrevoets et al. (2025) for differentiable causal discovery in nonlinear ANMs with arbitrary noise distributions and variances. We first argue that the global minimizer of any oracle independence criterion recovers the true underlying DAG, up to additional constant edges. In fact, this follows from the identifiability of ANMs, and therefore, we do not refer to the theory of independence component analysis (ICA) as Berrevoets et al. (2025), which poses a fundamental difference and difficulty in the nonlinear scenario due to its unidentifiability. The additional constant edges correspond to functionally trivial parent-child relationships and create a new challenge in the nonlinear setting. To address the problem, we introduce an adaptive group lasso penalty (Wang & Leng, 2008) that regularizes entire columns of the first-layer weight matrix in a neural representation of the ANM. This encourages the pruning of constant or uninformative components in a structure-aware manner, unlike standard elementwise $L_1$ penalties. We also propose a CAM-based initialization scheme to reduce the optimization burden of the resulting nonconvex objective.

**Related Works:** There are several additional works on differentiable causal structural learning using loss functions other than MSE and log-likelihood. Here we discuss their contributions and limitations. For example, DARING (He et al., 2021) also minimizes the joint dependence between the residuals in the loss function. However, the authors did not show that the minimizer corresponds to the underlying DAG, and actually, there could be some uninformative edges due to violation of causal minimality, see our Theorem 3.1. In addition, to ensure a consistent dependence measure, both of the residuals $R_{-i}$ and $R_i$ in their loss function need to be transformed by nonlinear functions. Nevertheless, they only transformed $R_{-i}$ due to the computational burden. In Chen et al. (2023), the authors proposed a loss function based on the entropy of the regression residuals. They showed a con-

nection between their loss function and the log-likelihood if the regression function is correctly specified. Moreover, minimizing the residual entropy yields bivariate identifiability. However, it remains unclear whether a consistent estimator can be obtained by minimizing the entropy loss in the *multivariate case*, see also Kpotufe et al. (2014). For example, in LiNGAM, the causal ordering is identified if and only if the residuals are mutually independent, i.e., have zero mutual information. According to Hyvärinen (1997), the mutual information of the residuals can be expressed as the sum of the residual entropy minus a log-determinant term. Therefore, it is unclear whether the last term can be dropped in the optimization. Moreover, when the regression function is wrong, e.g., by using incorrect parent variables, the sum of the log-likelihood is only equal to the sum of residual entropy *conditioned* on the (potentially wrong) parent variables, not the sum of the entropy itself. Therefore, provable multivariate identifiability of differentiable causal learning methods on nonlinear ANMs remains a question.

**Contributions:** On the contrary, our paper introduces a differentiable causal discovery method with theoretical identifiability guarantees for general (identifiable) nonlinear ANMs. In particular,

1. We theoretically show that by minimizing the joint dependence measure of the residuals , rather than conducting independence tests, we identify the true topological ordering in nonlinear ANMs on a population level. Moreover, we identify the true DAG up to some edges connecting to uninformative variables, i.e., the derivative of the underlying function with respect to them is zero given the true parents. We further provide finite-sample guarantees for the probability of recovering the true topological ordering of the DAG.

2. We introduce an efficient pipeline for minimizing residual dependence, using an adaptive group lasso penalty to remove uninformative edges and a CAM-based initialization scheme to improve optimization stability.

3. Empirical studies on synthetic and real-world datasets show that our method outperforms or is comparable to existing differentiable methods in accuracy. Moreover, it consistently shows high robustness to noise heterogeneity, which existing methods do not have.

## 2. Preliminaries

**Nonlinear Additive Noise Models (ANMs).** Let $X = (X_1, \ldots, X_d)^\top$ be a $d$-variate random vector. In this paper, we consider the nonlinear ANMs on $X$, where each variable $X_j$ is defined as

$$X_j = f_j(\mathbf{PA}_j) + \varepsilon_j. \tag{1}$$

Here $\mathbf{PA}_j \subset \{X_1, \ldots, X_d\} \setminus \{X_j\}$ denotes the set of parents of $X_j$, and $f_j$ is a nonlinear function. The noise terms $\boldsymbol{\varepsilon} = (\varepsilon_1, \ldots, \varepsilon_d)^\top$ are jointly independent and $\varepsilon_j \perp\!\!\!\perp \mathbf{PA}_j$ for all $j \in \{1, \ldots, d\}$. The structural assignments induce a directed acyclic graph (DAG) $\mathcal{G}_0$ on $\boldsymbol{X}$. It is well-known that the DAG underlying the ANM is identifiable using observational data assuming the restricted bivariate identifiability holds (Peters et al., 2014).

**Notations:** We denote the set of the descendants of $X_j$ by $\mathbf{DE}_j$. $\hat{\mathcal{G}}_n, \mathcal{G}_0, \mathcal{G}_0^{min}$ represent the estimated DAG based on $n$ datapoints, the DAG generated by the ANM, and the causal minimal DAG corresponding to the ANM. For two DAGs $\mathcal{G}_1, \mathcal{G}_2$, we say $\mathcal{G}_1 \geq \mathcal{G}_2$ if $\mathcal{G}_2$ is a subgraph of $\mathcal{G}_1$.

**Differentiable Causal Structure Learning on ANMs.** Differentiable DAG learning methods reformulate the combinatorial problem of structure discovery into a continuous optimization problem with a suitable loss function and an acyclicity constraint. The general form was initially proposed in the seminal paper by Zheng et al. (2018). Subsequent work has explored different DAGness constraints and optimization strategies (Ng et al., 2020; Wei et al., 2020; Bello et al., 2022; Zhang et al., 2025). In particular, it was shown that any polynomial function or analytic function on the positive weight matrix can be applied as a constraint. To deal with general nonlinear ANMs, a popular approach is to parameterize $f_j$ by neural networks (NN) $F_{\theta_j}$ (typically MLP) with parameters $\boldsymbol{\theta} = (\theta_1, \ldots, \theta_d), 1 \leqslant j \leqslant d$. The general optimization problem is thus defined as

$$\boldsymbol{\theta}^* = \underset{\boldsymbol{\theta}}{\operatorname{argmin}} L(\boldsymbol{X}, (F_{\theta_1}, \ldots, F_{\theta_d})) + \lambda g(\boldsymbol{\theta})$$
$$\text{s.t. } h(\boldsymbol{\theta}) = 0, \tag{2}$$

where $L$ is the loss function, $\lambda \geqslant 0$, $g(\boldsymbol{\theta})$ is the sparsity penalty on the NN architecture, and $h(\boldsymbol{\theta})$ is the DAG constraint, c.f. the trace-exponential constraint in NOTEARS (Zheng et al., 2018). Augmented Lagrangian and barrier methods are commonly used for parameter optimization. In practice, we can merge all the $d$ NNs into a single NN. For the DAG constraint on NN, GraN-DAG (Lachapelle et al., 2020) defines a connectivity matrix based on neural path activations and applies the acyclicity constraint to this matrix. Zheng et al. (2020) proved that the columns in the first layer $W^{(1)}$ of the NN contain all the information for the acyclicity and sparsity. Specifically, the class of functions that are independent of the $k$-th variable (thus $X_k$ should not be the causal parent) can be equivalently expressed by an NN whose $k$-th column is zero. Thus, it suffices to define $g$ and $h$ as functions of $W^{(1)}$. Compared to the approach of GraN-DAG, it is more computationally efficient.

**NOTIME.** Recently, Berrevoets et al. (2025) introduced a principled differentiable approach NOTIME for learning linear ANMs with non-Gaussian noise using an independence-

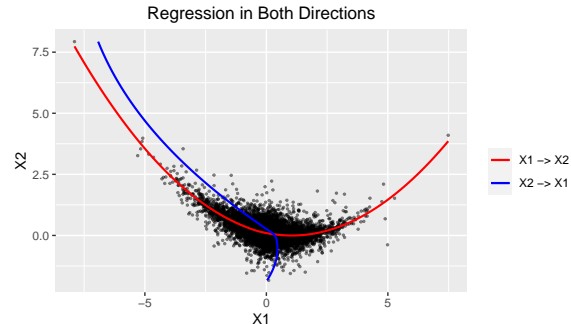

*Figure 1.* Regression functions for both causal directions on $X_1, X_2$. $X_1 \rightarrow X_2$ is the true direction.

based objective. Rather than relying on MSE or log-likelihood, NOTIME minimizes a multivariate dependence measure, e.g., d-variate Hilbert-Schmidt Independence Criterion (dHSIC) (Pfister et al., 2018) under a DAG constraint, and proves that the global minimizer corresponds to the true DAG. This framework serves as a foundation for extending identifiability guarantees of differentiable structure learning methods. Note that DARING (He et al., 2021) and Chen et al. (2023) also proposed alternative loss functions, but the multivariable identifiability of their methods was not clearly justified, see previous discussion of related works.

**Identifiability guarantees for learning ANMs.** The identifiability of ANMs implies a unique causally minimal DAG $\mathcal{G}_0^{min}$ that gives rise to the variable distribution. In other words, provided we have sufficient data, it should be possible to identify the exact DAG. Unfortunately, existing causal learning methods with theoretical guarantees for identifying the underlying DAG often fall short in scalability, c.f. (Bühlmann et al., 2014; Rolland et al., 2022; Peters et al., 2014). Differentiable DAG learning methods scale well with the dimension, but the commonly used MSE and log-likelihood-based loss functions are only theoretically proved to be consistent for noise with equal variance (Loh & Bühlmann, 2014) or prespecified distributions. For nonlinear ANMs, the condition for the MSE loss to be consistent is *not fully known* despite some empirical evidence (Reisach et al., 2021; Ng et al., 2024). Below, we provide an example showing that MSE can prefer the wrong causal order with statistically significant evidence in a bivariate nonlinear ANM with heterogeneous noise.

*Example* 2.1. We consider two variables defined by the equations, $X_1 = \varepsilon_1$, $X_2 = \beta(X_1 - 1)^2 + \varepsilon_2$, where $\beta = 0.1$, $\text{Var}(\varepsilon_1) = 1$, $\text{Var}(\varepsilon_2) = 0.1$, each from the Laplace distribution. We generate $n = 10000$ observations and calculate the MSE based on the hypothesis $X_1 \rightarrow X_2$, and $X_2 \rightarrow X_1$. We use the local polynomial estimator as the fitting function to prevent function class misspecification for the true direction (for the wrong direction, the MSE may be

further improved by choosing a suitable function class!). An illustration of the data and the regression functions is given in Figure 1. The procedure is repeated 100 times, and we calculate the mean and standard deviation of the MSE based on the hypotheses and the $p$-value resulting from the paired t-test. Table 1 shows that the wrong direction is always preferred by the MSE loss with a significant $p$ value. This confirms its inconsistency in nonlinear ANMs.

## 3. Identifiability by dependence measure

In this section, we extend the analysis in NOTIME by showing that any oracle dependence measure yields a consistent loss function for learning *nonlinear* ANMs in the differentiable framework. Moreover, we analyze the behavior of the global minimizer in finite-sample settings and prove the convergence of $P(\hat{\mathcal{G}}_n \geq \mathcal{G}_0^{min})$ to 1.

### 3.1. Dependence measures are consistent loss functions

As outlined in the introduction, causal discovery on nonlinear ANMs with general identifiability guarantees and scalability remains an open problem. Recently, Berrevoets et al. (2025) proposed NOTIME, which minimizes the dependence between the residuals by the d-variate Hilbert-Schmidt Independence Criterion (dHSIC) (Pfister et al., 2018) in the loss function, and achieves identifiability guarantees on LiNGAM. Since general ANMs also rely on the independent noise assumption, it is natural to ask whether the theoretical guarantees can be extended to this scenario. The original proof for NOTIME was based on the identifiability of linear ICA (Comon, 1994), a well-studied problem closely related to LiNGAM, which identifies the causal ordering up to permutation and rescaling of the weight matrix. The DAG constraint is then applied to determine the permutation matrix and prove the identifiability. Unfortunately, due to the unidentifiability of nonlinear ICA (Hyvärinen & Pajunen, 1999) and its unique functional form, the proof does not generalize to nonlinear ANMs. In the following, we argue that the approach of NOTIME can actually identify general ANMs up to some uninformative edges (Equation (4)). Our proof is inspired by the identifiability of ANMs, which rules out alternative causal structures with missing or reversed edges relative to the true causal minimal graph. We refer to Appendix B for the preliminary results, and Appendix C.1 for the proof.

*Table 1.* Results on MSE and paired t-test for two causal directions.

| Direction | MSE ± std | $p$-value | Errors |
|---|---|---|---|
| $X_1 \rightarrow X_2$ (True) | 1.101±0.0004 | <2.2e-16 | 100 |
| $X_2 \rightarrow X_1$ | 0.952±0.0006 | | |

**Theorem 3.1.** *Let $\boldsymbol{X} = (X_1, \dots, X_d)$ be the set of the vari-*

*ables generated by an identifiable ANM (e.g., by restrictive bivariate identifiability, c.f. Peters et al. (2014))*

$$X_j := f_j(\boldsymbol{PA}_j) + \varepsilon_j,$$

*for some nonlinear function $f_j$ and mutually independent noise $\varepsilon_j$. Let DM: $\mathbb{R}^d \rightarrow [0, \infty)$ be any oracle dependence measure on $d$ variables which is equal to zero if and only if the variables are mutually independent. Assume that each $f_j$ can be parameterized by $F_{\theta_j^*} : \mathbb{R}^d \rightarrow \mathbb{R}$ restricted to $\boldsymbol{PA}_j$, for some neural network parameter $\theta_j^* \in \Theta$. Let $\boldsymbol{\theta} = (\theta_1, \dots, \theta_p), \theta_j \in \Theta,$*

$$\hat{\boldsymbol{\theta}} = \underset{\boldsymbol{\theta}}{\arg\min}\, DM(X_1 - F_{\theta_1}(X), \dots, X_d - F_{\theta_d}(X))$$

$$s.t.\ h(\boldsymbol{\theta}) = 0,$$

$$(3)$$

*where $h(\boldsymbol{\theta})$ stands for a DAG condition on the parameters $\boldsymbol{\theta}$ (e.g., the trace-exponential condition on the first-layer weights $W^{(1)}$ (Zheng et al., 2020)). Then, on a population level, we have*

$$F_{\hat{\theta}_j}(\boldsymbol{X}) = F_{\hat{\theta}_j}(\boldsymbol{PA}_j, \boldsymbol{X} \backslash \boldsymbol{PA}_j) = f_j(\boldsymbol{PA}_j),$$

$$and\ \frac{\partial F_{\hat{\theta}_j}}{\partial X_i} = 0, \forall X_i \notin \boldsymbol{PA}_j. \qquad (4)$$

*Remark* 3.2. For LiNGAM, if the value of $f_j$ is constant with respect to $X_i$ (Equation (4)), then the coefficient of $X_i$ has to be zero, which means that $X_i$ would not be included in $\boldsymbol{PA}_j$. Therefore, the global minimizer exactly corresponds to the underlying causal minimal DAG with no additional edges. The above result is then a generalization of Theorem 1 in Berrevoets et al. (2025).

Theorem 3.1 indicates that by choosing a suitable dependence measure as the loss function $L$ in Equation (2), the global minimizer recovers a supergraph of the causal minimal graph, which can be reduced to it by pruning functionally constant edges., i.e. by removing variables such that $\frac{\partial F_{\hat{\theta}_j}}{\partial X_i} = 0$. In Section 4, we analyze the NN achitecture and propose a suitable sparsity penalty to efficiently prune uninformative edges. We further propose an initialization method to alleviate the difficulty in optimization, since it remains an open problem in the literature of differentiable causal discovery whether the global minimum of the objective function could be attained by arbitrary initialization.

### 3.2. Finite-sample guarantees

The dependence of the variables can only be accessed by an estimator based on the observed data. In this paper, we continue to use dHSIC as in NOTIME to quantify dependence, which has proven to work well in practice. The estimator

on finite samples is

$$\widehat{\text{dHSIC}}(\boldsymbol{x}^1, \ldots, \boldsymbol{x}^d) = \frac{1}{n} \sum_{M_2(n)} \prod_{j=1}^d k^j(x_{i_1}^j, x_{i_2}^j)$$

$$+ \frac{1}{n^{2d}} \sum_{M_{2d}(n)} \prod_{j=1}^d k^j(x_{i_{2j-1}}^j, x_{i_{2j}}^j) \quad (5)$$

$$- \frac{2}{n^{d+1}} \sum_{M_{d+1}(n)} \prod_{j=1}^d k^j(x_{i_1}^j, x_{i_{j+1}}^j),$$

where $\boldsymbol{x}^j \in \mathbb{R}^{n \times 1}$ is the $j$-th variable, $k^j$ is the kernel function for $\boldsymbol{x}^j$, and $M_q$ is the $q$-fold cartesian product on the samples with size $n$. It is therefore necessary to investigate the finite-sample property of the *global minimizer* (based on the $\widehat{dHSIC}$) to understand how fast it would converge to the true causal minimal DAG $\mathcal{G}_0$, for which we need Assumption 3.3.

**Assumption 3.3.** We assume the following hold:

1. The estimator $\hat{f}_j(X_S) = \hat{E}[X_j | X_S]$ converges uniformly to some $f_j^*(X_S)$ in probability with the rate $r(n) \to 0$, for any $X_j \in \{X_1, \ldots, X_d\}$ and $X_S \subset \{X_1, \ldots, X_d\} \backslash \{X_j\}$, i.e. $\|\hat{f} - f^*\|_2 = O_p(r(n))$. Moreover, $f_j^*(X_S)$ only needs to be $E[X_j | X_S]$, i.e. consistent, if $\mathbf{PA}_j \subset X_S \subset \{X_1, \ldots, X_d\} \backslash \{X_j, \mathbf{DE}_j\}$.

2. The function class $\hat{\mathcal{F}}$ s.t. $\hat{f} \in \hat{\mathcal{F}}$ and the class of noise distributions satisfy Condition 19 of (Peters et al., 2014), i.e. restricted bivariate identifiability.

3. The kernel function in dHSIC is bounded and Lipschitz continuous.

The last two assumptions are standard for the identifiability of ANMs and kernel-based dependence measures. For the first assumption, the general idea is that the convergence of the regression functions induces the convergence of the residuals. Therefore, we can approximate the theoretical statistic $dHSIC(\mathbb{P}_{\boldsymbol{\varepsilon}})$ based on the distribution of $\boldsymbol{\varepsilon}$ by the estimates $\widehat{dHSIC}(\hat{\varepsilon}_1, \ldots, \hat{\varepsilon}_d)$. Note that many estimators satisfy the first assumption. For example, if the class of squared-loss functions (induced by the class of regression functions) is Glivenko–Cantelli (Van Der Vaart & Wellner, 1996), then the empirical risk converges uniformly to the population risk. If additionally the population risk has a well-separated minimizer $f^*$ in the sup-norm topology, then standard argmin-consistency arguments imply $\|\hat{f} - f^*\|_\infty = o_P(1)$. We provide further discussions in Remark C.1 in Appendix C.

**Corollary 3.4.** *Let $\hat{\varepsilon}_j = X_j - \hat{f}(X_{S_j})$, $\varepsilon_j = X_j - f^*(X_{S_j})$ for some parent set $S_j$ of $X_j$ induced by a causal graph and $\mathbb{P}_{\boldsymbol{\varepsilon}}$ be its distribution. Under Assumption 3.3, we have*

1. *$\widehat{dHSIC}(\hat{\varepsilon}_1, \ldots, \hat{\varepsilon}_d) \xrightarrow{P} dHSIC(\mathbb{P}_{\boldsymbol{\varepsilon}})$ with the rate $O_P(r(n) + n^{-1/2})$.*

2. *Any global minimizer of Equation (3) based on $n$ datapoints and $DM = \widehat{dHSIC}$ induces a DAG $\hat{\mathcal{G}}_n$ s.t.*

$$P(\hat{\mathcal{G}}_n \geq \mathcal{G}_0) = 1 - o(1).$$

The proof is deferred to Appendix C.2. Note that the result can be generalized to any consistent and Lipschitz continuous dependence measure. A similar finite-sample result for identifying *bivariate* ANMs based on differential entropy was proved in Kpotufe et al. (2014). However, further conditions on the consistency of $\hat{f}$ in the anti-causal direction and the variable distribution have to be assumed.

## 4. Methodology

In this section, we introduce our learning method NOTIME-CAM, which incorporates an adaptive group lasso penalty and CAM-based initialization, using the dHSIC dependence measure as the training objective.

### 4.1. Adaptive Group Lasso Penalty

To induce sparsity in the estimated DAG, we employ an adaptive group lasso penalty on the first-layer weights of the neural networks $\{F_{\theta_j}\}_{j=1}^d$, where each $F_j$ models the structural function for $X_j$ given its parents. Unlike standard $L_1$ penalties adopted in NOTEARS-MLP and DAGMA, which act independently on each parameter, our penalty recognizes that the existence of an edge $i \to j$ is determined by the entire $i$-th input column of the first layer of $F_{\theta_j}$. Therefore, we penalize the $L_2$ norm of each input column of the first layer $W_j^{(1)} \in \mathbb{R}^{m_1 \times d}$ of network $F_{\theta_j}$ (with hidden units $m_1$) as a group. This encourages the removal of all weights associated with non-parent variables and ensures edge-level sparsity that is functionally meaningful. Specifically, we define the group lasso penalty as:

$$g(W^{(1)}) = \sum_{j=1}^d \sum_{i \neq j} \omega_{ij} \left\| W_j^{(1)}[:, i] \right\|_2, \quad (6)$$

where $W_j^{(1)}[:, i] \in \mathbb{R}^{m_1}$ denotes the $i$-th column of $W_j^{(1)}$, $W^{(1)}$ is the first layer in the merged neural network. To further improve support recovery, we adopt adaptive reweighting of the group norms, which is defined by setting $\omega_{ij}$ in Equation (6) equal to the following:

$$\omega_{ij} = \frac{1}{\left\| \hat{W}_j^{(1)}[:, i] \right\|_2^\beta + \epsilon}, \quad (7)$$

with a preliminary estimate of the coefficients $\hat{W}_j^{(1)}$ (which we will define in the next subsection), a tuning parameter $\beta > 0$ for the scale (typically set to 0.5, 1, or 2) (Wang & Leng, 2008; Dinh & Ho, 2020), and a small constant $\epsilon > 0$ for numerical stability. This reweighting scheme assigns a *lower penalty to strong signals* and a *higher penalty to weak or irrelevant inputs*. This is crucial when the effect sizes of different parent variables vary significantly or when variables have differing scales, which is common in practice. Uniform penalties tend to shrink large and small coefficients indiscriminately, which can lead to biased estimation and false exclusions of weaker but important signals. In contrast, adaptive group lasso mitigates these issues by assigning group-specific penalties. Theoretically, adaptive lasso-type methods are known to achieve oracle properties, recovering the correct support with asymptotically unbiased estimates under suitable conditions (Zou, 2006; Wang & Leng, 2007). Our adaptive group lasso thus offers both statistical and structural advantages tailored to the architecture and goals of neural DAG learning. This makes our learning target:

$$\boldsymbol{\theta}^* = \underset{\boldsymbol{\theta}}{\text{argmin}}\,\widehat{\text{HSIC}}(X_1 - F_{\theta_j}(X), \ldots, X_d - F_{\theta_d}(X))$$
$$+ \lambda g(W^{(1)}) \quad \text{s.t. } h(\boldsymbol{\theta}) = 0,$$

where the group lasso penalty is defined as Equations (6) and (7), $h$ is the DAG constraint for which we follow NOTEARS-MLP (Zheng et al., 2020). Note that NOTIME also pursued the same idea by de-standardizing the weight by a matrix $\Sigma$ before applying the regularization to make the coefficients less dependent on the scale of the variables. However, this approach may not be suitable for nonlinear, non-additive relationships.

### 4.2. Initialization and the Adaptive Penalty

As indicated by Ng et al. (2024); Berrevoets et al. (2025), initialization is crucial for the performance of differentiable learning methods due to the complex landscape of the loss functions. Therefore, we seek an initialization whose weight matrices and induced graph are already close to a good solution, thereby reducing the optimization burden. With such a good initialization point, the weight matrices could also provide information on the adaptive penalty to guide sparsity regularization. Note that to achieve this goal, the initialization method has to provide certain identifiability guarantees on the model.

For LiNGAM, the authors of NOTIME showed that initialization by ICA-LiNGAM was successful. For nonlinear ANMs, however, it is not straightforward to have a good initialization of the NN weight matrices. For example, we cannot initialize our model with the result of NOTEARS or DAGMA using the MSE loss due to its inconsistency in general settings. Combinatorial optimization methods, on the other hand, have shown robust performance to data

standardization (Rolland et al., 2022) but do not use NN as the fitting function. While some existing methods (Nazaret et al., 2024) use a partial directed moral graph, e.g., PC (Colombo et al., 2014) to mask out edges that are not discovered by constraint-based methods for sparsity control and skeleton initialization, they are not purely informative on the value of weight matrices. This is because the fitting function for a response variable can be totally different when including both causal and anti-causal predictors.

To address this issue, we propose a mechanism that allows stable initialization from CAM and an adaptive group Lasso penalty. Specifically, we first fit a CAM on the data to extract the predicted DAG. Then we restrict the columns in the first layer $W_j^{(1)}$ of our NN by a 0-1 mask $\mathcal{M}_{\text{order}}$ using the causal ordering from CAM, i.e. the $i$-th column of $W_j^{(1)}$ is fixed to be zero if and only if the causal order of $X_j$ comes before $X_i$. In the second step, we train the NN with the mask using the MSE loss to learn the prediction function given the causal ordering. Then, we calculate the $L_2$ norm of the columns in the first layer and use it as the adaptive penalty in Equation (7). In order to mitigate the possible error resulting from learning the skeleton, we add a threshold $\text{thr\_adapt} > 0$ to the denominator of $\omega_{ij}$ (adaptive coefficients) for edges that do not appear in CAM, i.e., either not in the skeleton, or reversed. Otherwise, these edges would have adaptive coefficients as large as $1/\epsilon$ and therefore never be considered in the learning procedure. In the next step, we define another 0-1 mask $\mathcal{M}_{\text{DAG}}$ to mask out all the columns that are not in the DAG of CAM, i.e. the $i$-th column of $W_j^{(1)}$ is fixed to be zero if and only if there is no edge from $X_i$ to $X_j$. In other words, $\mathcal{M}_{\text{order}}$ is based on the causal ordering while $\mathcal{M}_{\text{DAG}}$ is based on the DAG prediction. Finally, we fix other layers in the NN, and retrain the first layer $W_j^{(1)}$ based on $\mathcal{M}_{\text{DAG}}$ to obtain an NN that induces the same DAG structure of CAM and minimizes the MSE loss. This becomes our initialization from CAM. Details are summarized in Algorithm 1.

*Remark* 4.1. Note that the use of MSE loss in the initialization part does not contradict our claim that it is not consistent for general ANMs. We use the MSE loss when the causal ordering is first estimated in a relatively reasonable way (c.f. Rolland et al. (2022)). In this case, the parent selection problem can be reduced to the variable selection problem in regression (Bühlmann et al., 2014; Deng et al., 2023). Thus, we can minimize the MSE to efficiently find the solution. In the framework of differentiable DAG learning, however, the goal is to minimize the MSE *globally* among all possible causal orderings and prediction functions. For this, Example 2.1 has shown that the wrong causal ordering may lead to a smaller MSE than the ground truth.

*Remark* 4.2. The reason why we first learn the adaptive coefficients and then the sparse prediction function is that neural networks are flexible enough to have various parame-

---

**Algorithm 1** NOTIME-CAM initialization

---

**Require:** Data matrix $\boldsymbol{X} \in \mathbb{R}^{n \times d}$, $\eta > 0$ for thr_adapt, $\beta > 0$

**Ensure:** Initialized weights $W^{(1)} = (W_1^{(1)}, \ldots, W_d^{(1)})$, adaptive group lasso coefficients $\omega_{ij}$

 1: **Step 1: Fit CAM and define ordering mask**
 2: Fit CAM on $\boldsymbol{X}$ to get causal order $\pi$ and estimated DAG $\widehat{\mathcal{G}}$
 3: Construct mask $\mathcal{M}_{\text{order}} \in \{0,1\}^{d \times d}$ where $\mathcal{M}_{\text{order}}[i,j] = 1$ if and only if $\pi(i) < \pi(j)$
 4: **Step 2: MSE pretraining with $\mathcal{M}_{\text{order}}$**
 5: Apply $\mathcal{M}_{\text{order}}[:,j]$ to mask columns in $W_j^{(1)}$
 6: Train the NN with MSE loss, keeping masked entries fixed to zero
 7: **Step 3: Compute adaptive penalty coefficients**
 8: **for** $j = 1$ to $d$ **do**
 9:   **for** $i = 1$ to $d$ **do**
10:     **if** $i \neq j$ **then**
11:       $\ell_{ij} \leftarrow \left\| W_j^{(1)}[:,i] \right\|_2$
12:       **if** $(i \rightarrow j) \in \widehat{\mathcal{G}}$ **then**
13:         $\omega_{ij} \leftarrow 1/(\ell_{ij} + \epsilon)$
14:       **else**
15:         $\omega_{ij} \leftarrow 1/(\ell_{ij} + \text{thr\_adapt} + \epsilon)$
16:       **end if**
17:     **end if**
18:   **end for**
19: **end for**
20: **Step 4: Retrain $W^{(1)}$ with sparse mask**
21: Construct $\mathcal{M}_{\text{DAG}} \in \{0,1\}^{d \times d}$ where $\mathcal{M}_{\text{DAG}}[i,j] = 1$ if and only if $(i \rightarrow j) \in \widehat{\mathcal{G}}$
22: Apply $\mathcal{M}_{\text{DAG}}$ to $W^{(1)}$, freeze other layers
23: Retrain $W^{(1)}$ with MSE loss

---

ters that minimize the MSE loss. Therefore, we want to stay in a neighbourhood where the coefficient of the adaptive penalty is valid. To achieve this, we fix all the other layers and only fine-tune the first layer where we impose an extra sparsity constraint in Step 4 of Algorithm 1.

**Optimization.** We use the augmented Lagrangian method and the trace-exponential DAG constraint as NOTEARS and NOTIME, we refer to their papers for more details of the optimization algorithm. We note that it is possible to incorporate our method into other optimization methods, such as the barrier method used in DAGMA. But we did not observe a significant computational advantage.

## 5. Experiments

In this section, we compare our method with baseline methods on synthetic and real-world datasets that are generated by various noise distributions and data scaling schemes.

**NOTIME-CAM.** NOTIME-CAM uses the same main hyperparameters as NOTIME, except for three additional parameters $\beta, \epsilon$, and thr_adapt for constructing the adaptive group lasso penalty. In our experiments, we set thr_adapt to be $\eta = 0.1$ times the 10% quantile of the $\omega_{ij}$ in line 12 of Algorithm 1 to make it data-driven. This parameter is intended to allow potential edges not identified by CAM to enter the model, so it should be neither zero nor too large. $\epsilon = 1e^{-12}$ is a stabilizer for the gradient, and $\beta = 0.5$ to regularize more on large weights. Steps 2, 4 use the Adam optimizer with $lr = 0.1$. For the loss function $\widehat{dHSIC}$, we use the Gaussian kernel and median bandwidth. We provide our code at https://github.com/STAN-UAntwerp/NOTIME.

### 5.1. Synthetic datasets

**Simulation Setup.** Our setup is aligned with Zheng et al. (2020); Bello et al. (2022) on simulating nonlinear DAGs. We simulate datasets of size $n = 1000$ with $d \in \{10, 30, 50, 70\}$ variables. We use Erdös-Renyi (ER) graphs with degree 4, mainly to consider dense graphs. The functions that generate the data are two-layer MLPs with a hidden dimension $m_1 = 100$ and the sigmoid activation function. The weights are sampled randomly from a uniform distribution on $[-2, -0.5] \cup [0.5, 2]$. For the noise variables, we consider three different error distributions: standard Gaussian distribution, lognormal distribution with scale parameter 1, and $t(3)$ distribution. Note that the mean is subtracted for the lognormal distributions. Finally, we fit a two-layer MLP which has hidden units $m_1 = 10$ on the original data, the data scaled to equal variances, and the data scaled so that the marginal variances of the variables are in the reverse order as Berrevoets et al. (2025). The non-Gaussian noise distributions and scaling schemes are designed to assess both identifiability and robustness of NOTIME-CAM and the baseline methods.

**Baseline Methods and Parameter Settings.** We compare our method with CAM, NOTEARS-MLP, DAGMA, and GraN-DAG. In particular, CAM is used for initialization and has shown strong performance in recent studies (Rolland et al., 2022). We use the implementation from Kalainathan et al. (2020) with a default cutoff parameter of 0.001. The latter three methods are the state-of-the-art (SOTA) differentiable structure learning methods which use different DAG constraints for which we use the implementation from the authors. For NOTEARS-MLP, DAGMA, and NOTIME-CAM, we consider both the default parameters and the oracle hyperparameters for the (group) Lasso penalty, and the cutoff threshold. We did not compare with DARING (He et al., 2021) as the code is not publicly available. For evaluation, we report the structural Hamming distance (SHD) of the predictions, the standard deviation of

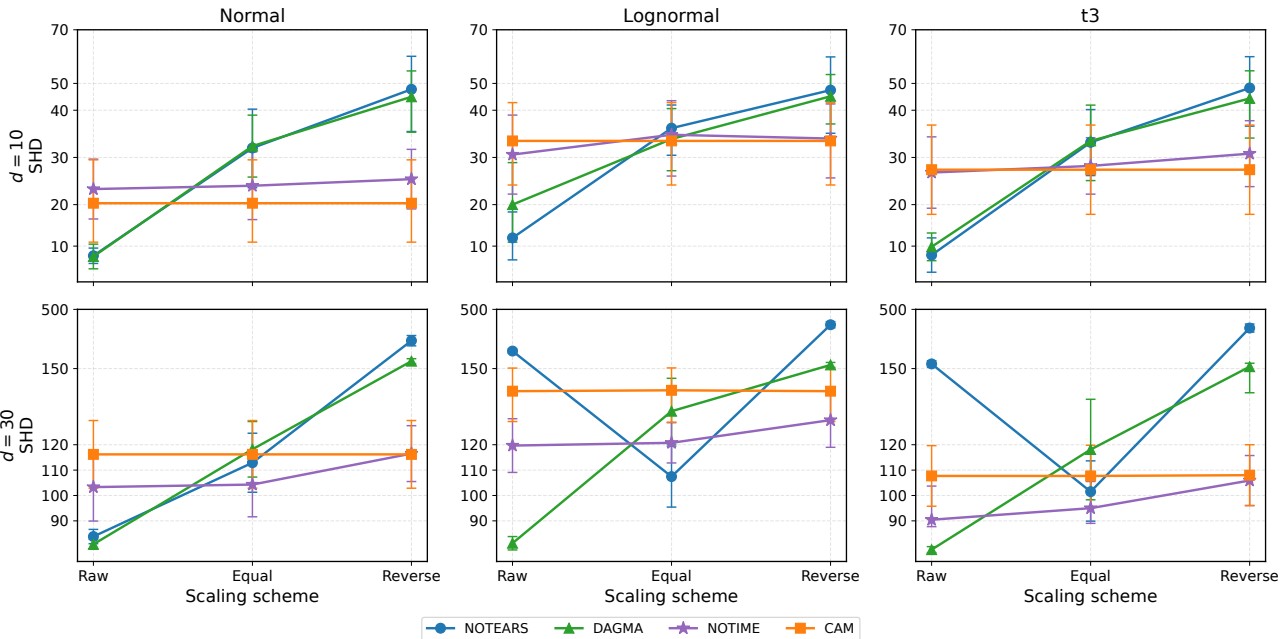

*Figure 2.* Mean SHD (lower better) to the ground truth for $d = 10, 30$ with default hyperparameters. From left to right: Results for normal, lognormal, and $t(3)$ noise distribution. X-axis: Scaling scheme. Y-axis: SHD. Error bars based on the standard deviation.

SHD, TPR, and FDR (see Appendix E.1). More details on the hyperparameters are provided in Appendix D.

**Results.** Figures 2 and 3 show the SHD results for the default hyperparameters. The results for the unscaled (raw) data are only for reference, as the setting favors the MSE loss by design (Reisach et al., 2021). It is clear that NOTIME-CAM is robust across three data scaling schemes and noise distributions, and consistently improves over CAM. In contrast, NOTEARS and DAGMA are sensitive to the scaling schemes. This again highlights the importance of having a consistent loss function. More importantly, NOTIME-CAM is almost always the best method when the noise distribution is normal and $t(3)$. On lognormal noise, it outperforms DAGMA while being comparable to NOTEARS. Figure 6 in Appendix E shows the results for the oracle hyperparameters that are unavailable in practice. NOTIME-CAM remains the best method when the noise distribution is $t(3)$ or the data is reversely scaled (NOTEARS and DAGMA only output zero graphs!). For other settings, they are comparable. We note that GraN-DAG requires much more computational resources than other differentiable methods, and is more unstable for non-Gaussian noise (see Figure 7 and Appendix D). Therefore, their results are excluded here.

**Runtime comparisons.** Table 2 calculates the median of the runtime $\pm$ IQR/2 for $d = 10, 50$. It shows that the runtime for training is comparable to the runtime of CAM for moderate dimensions such as $d = 50$. We argue that the computational cost is worth the identifiability guarantees,

which most existing loss functions do not provide.

### 5.2. Real-world dataset

To evaluate the performance of NOTIME-CAM on real-world problems, we compare it with CAM, NOTEARS and DAGMA on the well-known causal discovery benchmark "Sachs" (Sachs et al., 2005), with $n = 853$ samples, $d = 11$ variables, and 17 known edges. The results are given in Table 3. We see that NOTIME-CAM clearly outperforms all the competitors. For default parameters, NOTIME-CAM predicted 5 edges with $SHD = 15$. We also provided the results for the hyperparameters giving the lowest error. The optimal parameter for NOTIME-CAM is $\lambda_1 = 2, \lambda_2 = 0.001$, with a hard cutoff_thr $= 0.1$, under which it has $SHD = 14$ and predicted 3 edges.

More importantly, CAM performs poorly with $SHD = 30$. Therefore, the experiment clearly shows that (1) CAM itself can perform poorly and (2) even when NOTIME-CAM is initialized with a relatively poor CAM fit, it can still substantially improve it, which highlights the utility of optimization under the independence measure.

*Table 2.* Comparison of the running time.

| $d$ | $t_{CAM}$ | $t_{init}$ | $t_{train}$ |
|---|---|---|---|
| 10 | $78.79 \pm 7.86$ | $1.53 \pm 0.33$ | $396.37 \pm 177.21$ |
| 50 | $721.22 \pm 38.65$ | $4.18 \pm 0.25$ | $676.03 \pm 576.66$ |

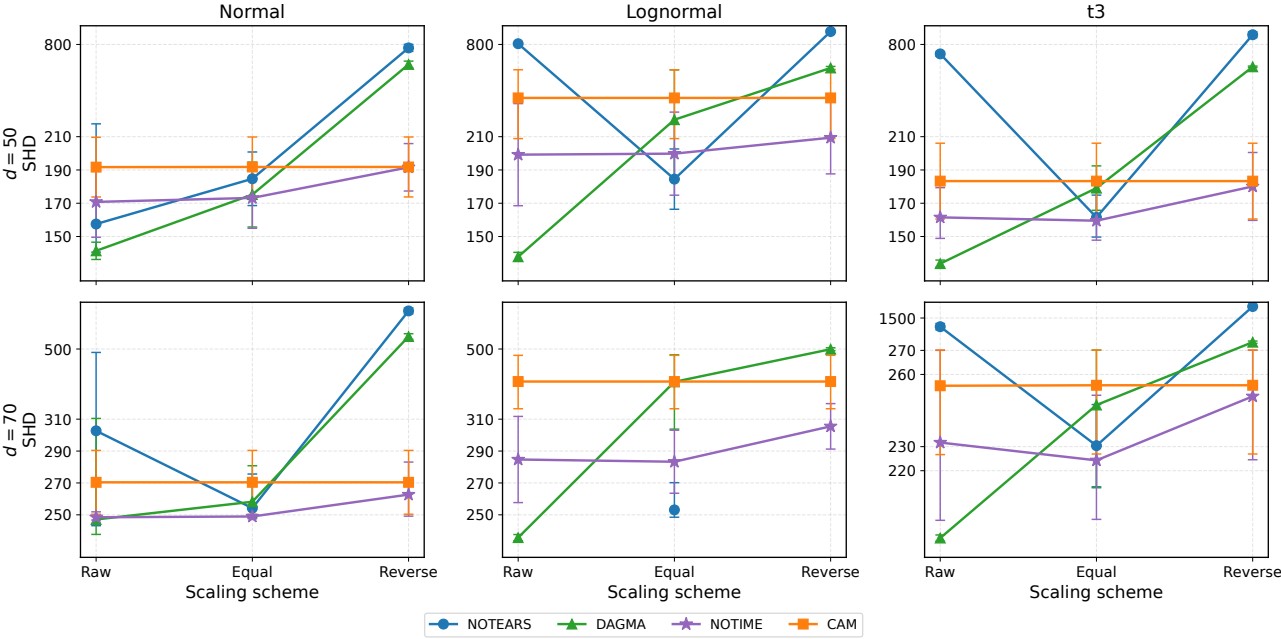

*Figure 3.* Mean SHD (lower better) to the ground truth for $d = 50, 70$ with default hyperparameters. From left to right: Results for normal, lognormal, and $t(3)$ noise distribution. X-axis: Scaling scheme. Y-axis: SHD. Error bars based on the standard deviation.

### 5.3. Additional experiments

**Weak signals.** We further conducted experiments with hidden units $m_1 = 10$ in the data generation mechanism so that CAM performs even worse. Figure 7 shows the results for the oracle hyperparameters (including GraN-DAG). The results are similar to Figures 2, 3 and 6.

**Ablation studies.** In Table 5, we compare our proposal with 3 variants as an ablation study: (1) Default: $L_1$ penalty with random initialization. (2) glasso: group lasso penalty with random initialization. Since we do not have an initial guess of the parameters (from CAM initialization in our proposal), we could not use the adaptive version. (3) CAM initialization with $L_1$ penalty. We see that no variant dominates the others in $d = 10$ and they all perform worse than our proposal, showing its effectiveness.

*Table 3.* Comparison of SHD, FDR, and TPR under default and optimal parameter settings.

| Method | Parameters | SHD | FDR | TPR |
|---|---|---|---|---|
| CAM | Default | 30 | 0.78 | 0.41 |
| **NOTIME-CAM** | Default | **15** | 0.4 | 0.18 |
| NOTEARS | Default | 22 | 0.76 | 0.29 |
| DAGMA | Default | 17 | 0 | 0 |
| **NOTIME-CAM** | Oracle | **14** | 0 | 0.18 |
| NOTEARS | Oracle | 16 | 0.7 | 0.17 |
| DAGMA | Oracle | 16 | 0 | 0.06 |

**Tuning thr_adapt.** In Table 6, We conducted experiments with thr_adapt be the 10% and 50% quantile of $\ell_{ij}$ times a constant $\eta$ for normal noise for different $d$ and computed the SHD in the table. We see that the default value works well, and in general our method is robust to small thr_adapt.

## 6. Conclusion

We propose a general method for causal discovery in nonlinear ANMs with arbitrary noise distributions and variances. By minimizing a differentiable independence-based loss, we prove that its global minimizer recovers the causal minimal DAG up to uninformative constant edges. A key component of our method is a novel CAM-based initialization scheme, which leverages a coarse causal ordering from the CAM algorithm to construct a masked neural architecture. This warm start both accelerates convergence and improves stability, particularly in high-dimensional or nonconvex regimes. We further impose an adaptive group lasso penalty on first-layer input columns, inducing edge-level sparsity that better matches the functional structure of neural DAG models than elementwise regularization.

Our method unifies the strengths of independence-based objectives and continuous DAG parameterizations, enabling scalable and consistent estimation across different noise types. Empirical results demonstrate stable and accurate recovery under challenging conditions where prior methods fail. Future work will focus on extending this framework to handle latent confounding and partial observability.

# Acknowledgements

This research received funding from the Flemish Government under the "Onderzoeksprogramma Artificiële Intelligentie (AI) Vlaanderen" programme (RY, TV). The work of Ruicong Yao was supported by Fonds Wetenschappelijk Onderzoek - Vlaanderen (FWO) through the research project V463924N. The work of Jakob Raymaekers was supported by Fonds Wetenschappelijk Onderzoek - Vlaanderen (FWO) through the research project G063626N.

# Impact Statement

This work contributes to more reliable causal discovery from observational data, particularly in settings where interventions are costly, infeasible, or ethically constrained. By improving identifiability and robustness for nonlinear additive noise models with heterogeneous noise, the proposed method can support scientific hypothesis generation in domains such as biology, medicine, economics, and policy analysis.

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

# A. Further discussions and limitations

**Limitations.** The ANM assumption has an additive structure between the noise and the variable, which allows estimation of independent residuals. This is crucial for the evaluation of the loss function in our method. Additionally, our theoretical results rely on the requirement that the global minimizer of the oracle dependence measure can be identified. However, we do not guarantee that any particular optimization method will find the global minimum without suitable initialization methods, which is also a common limitation for existing methods. Lastly, existing dependence measures often lose power for very high-dimensional data, which limits the application of our method to very large $d$, see also the discussions in Berrevoets et al. (2025).

# B. Preliminary results

In this section, we review some fundamental definitions and results in causal discovery, which help to prove Theorem 3.1 in Section 3.

**Definition B.1** (Causal Minimality (Peters et al., 2014)). A distribution satisfies causal minimality with respect to $\mathcal{G}$ if it is Markov with respect to $\mathcal{G}$, but not to any proper subgraph of $\mathcal{G}$.

Causal minimality restricts the graphical representations of the underlying distribution. That is, the edges in the graph can not be further reduced while satisfying the distribution and Markov property. This is a crucial property if we want to identify a unique graph consistent with the distribution. For ANMs, causal minimality is equivalent to a simple condition on the underlying function $f_j$.

**Proposition B.2** (Proposition 17 of Peters et al. (2014)). *Consider a distribution generated by an ANM and assume that the functions $f_j$ are not constant in any of their arguments, i.e., for all $j$ and $i \in \boldsymbol{PA}_j$ there are some $x_{\boldsymbol{PA}_j \setminus i}$, (i.e., the parent variables except $i$), and some $x_i \neq x_i'$ such that*

$$f_j(x_{\boldsymbol{PA}_j \setminus i}, x_i) \neq f_j(x_{\boldsymbol{PA}_j \setminus i}, x_i').$$

*Then the joint distribution satisfies causal minimality with respect to the corresponding graph. Conversely, if there is a $j$ and $i$ such that $f_j(x_{\boldsymbol{PA}_j \setminus i}, \cdot)$ is constant, causal minimality is violated.*

The next two results together show that for an identifiable ANM, there is a unique causal minimal graph with an associated ANM that leads to the underlying distribution $P(\boldsymbol{X})$, and $P(\boldsymbol{X})$ is Markov with respect to this graph.

**Lemma B.3.** *[Lemma 32 of Peters et al. (2014)] Let the distribution of the variables $P(\boldsymbol{X})$ be generated by an additive noise model with graph $\mathcal{G}_0$.*

1. *For each supergraph $\mathcal{G} \geq \mathcal{G}_0$, there is an additive noise model that leads to the distribution $P(\boldsymbol{X})$.*

2. *For each subgraph $\mathcal{G} \leq \mathcal{G}_0$ such that $P(\boldsymbol{X})$ is Markov with respect to $\mathcal{G}$ there is an additive noise model that leads to the distribution $P(\boldsymbol{X})$. Furthermore, there is an additive noise model with a unique graph $\mathcal{G}_0^{min} \leq \mathcal{G}_0$ that leads to $P(\boldsymbol{X})$ and satisfies causal minimality.*

**Corollary B.4** (Corollary 33 of Peters et al. (2014)). *Let the distribution $P(\boldsymbol{X}) = P(X_1, ..., X_p)$ be generated by an additive noise model with graph $\mathcal{G}_0$. Assume that the SEM corresponding to the minimal graph $\mathcal{G}_0^{min}$ defined as in Lemma B.3 is a restricted additive noise model. Consider an ordering $\pi$ and a restricted ANM with corresponding graph $\mathcal{G}_\pi^{full,min}$ that generates the distribution $P(\boldsymbol{X})$. Then the identifiability of the restricted additive noise model implies that $\mathcal{G}_\pi^{full,min} = \mathcal{G}_0^{min}$. In this sense, the set of true orderings*

$$\Pi_0 := \{\pi | \mathcal{G}_\pi^{full,min} \geq \mathcal{G}_0^{min}\}$$

*is identifiable from $P(\boldsymbol{X})$.*

# C. Proofs and discussions

## C.1. Theorem 3.1

*Proof of Theorem 3.1.* By the identifiability of nonlinear ANMs, we first have that there is a unique causal minimal graph $\mathcal{G}_0^{min}$ with an associated ANM that generates the data distribution. Using the DAGness constraint, the training result based

on our loss function is always a DAG (denoted by $\mathcal{G}$), which induces a causal ordering. By Corollary B.4, the true causal ordering is also identifiable from the data. This means that any ANM that agrees with the data distribution based on nonlinear functions $f_j$ and mutually independent noise would induce a graph that has the correct causal ordering, which is the case for our $\mathcal{G}$, given an infinite amount of data. Suppose there is an edge in $\mathcal{G}_0^{min}$ that is reversed in $\mathcal{G}$, then these two graphs induce different causal orderings which is not possible. Suppose on the other hand that $\mathcal{G}_0^{min}$ has an edge which is missing in $\mathcal{G}$, then by Lemma B.3, there exists another causal minimal graph $\mathcal{G}'^{min}_0$ which is a subgraph of $\mathcal{G}$ and there is an ANM with respect to it. Clearly $\mathcal{G}'^{min}_0 \neq \mathcal{G}_0^{min}$, and this contradicts our first sentence. Therefore, we can conclude that $\mathcal{G}$ must be a supergraph of $\mathcal{G}_0^{min}$ with some additional directed edges. These edges can be pruned to produce the causal minimal graph. Now, Proposition B.2 implies that if the causal minimality is violated for the edge $i \rightarrow j$ then $F_{\theta_j}$ is independent of $X_i$, which is equivalent to zero derivative. This completes the characterization of $F_{\hat{\theta}_j}$. $\qquad\square$

### C.2. Corollary 3.4

*Proof of Corollary 3.4.* We prove the results in three steps. The first two steps are direct by combining the reasonings in (Pfister et al., 2018). The third step is based on the union bound argument, with an illustration in Figure 4. Note that we regard the dimension $d$ as a fixed parameter.

1. We first have that $\widehat{dHSIC}$ and the empirical U-statistics of $dHSIC$ only defer in $O_p(1/n)$ uniformly by Lemma C.2 of (Pfister et al., 2018). Then, the U-statistics of $dHSIC$ converges to the theoretical statistics of $dHSIC$ in $o_P(n^{-1/2})$ by Theorem A from Section 5.6 of (Serfling, 2009). Thus,

$$\widehat{dHSIC}(\varepsilon_1, \ldots, \varepsilon_d) - dHSIC(\mathbb{P}_{\varepsilon}) = o_P(n^{-1/2}).$$

2. As the regression estimator converges uniformly, we have that for all possible parent sets $(S_1, \ldots, S_d)$,

$$||(\hat{\varepsilon}_1 - \varepsilon_1, \ldots, \hat{\varepsilon}_d - \varepsilon_d)||_2^2 = ||(\hat{f}(X_{S_1}) - f^*(X_{S_1}), \ldots, \hat{f}(X_{S_d}) - f^*(X_{S_d}))||_2^2$$
$$= \sum_{j=1}^{d} ||\hat{f}(X_{S_j}) - f^*(X_{S_j})||_2^2$$
$$= O_P(r^2(n))$$

Since the kernel function is Lipschitz and bounded, so is $\widehat{dHSIC}$ (Lemma E.1 of (Pfister et al., 2018)) in the sense that

$$|\widehat{dHSIC}(\hat{\varepsilon}_1, \ldots, \hat{\varepsilon}_d) - \widehat{dHSIC}(\varepsilon_1, \ldots, \varepsilon_d)| = O(||(\hat{\varepsilon}_1 - \varepsilon_1, \ldots, \hat{\varepsilon}_d - \varepsilon_d)||_2)$$

Combining the above two steps gives us

$$\widehat{dHSIC}(\hat{\varepsilon}_1, \ldots, \hat{\varepsilon}_d) - dHSIC(\mathbb{P}_{\varepsilon}) = O_P(r(n) + n^{-1/2}).$$

3. For any possible set of parents $\boldsymbol{S} = (S_1, \ldots, S_d)$, we now denote the distribution of the residuals $(\varepsilon_1^{\boldsymbol{S}}, \ldots, \varepsilon_d^{\boldsymbol{S}})$ by $\mathbb{P}_{\varepsilon}\boldsymbol{s}$ for clarity. Since $dHSIC(\mathbb{P}_{\varepsilon}\boldsymbol{s}) = 0$ if and only if the residuals are independent, the second condition of Assumption 3.3 implies that this can occur only for $\boldsymbol{S}$ whose induced DAG $\mathcal{G}_{\boldsymbol{S}}$ is a supergraph of the causal minimal graph $\mathcal{G}_0$ associated with the ANM, i.e.

$$dHSIC(\mathbb{P}_{\varepsilon}\boldsymbol{s}) = 0 \Longleftrightarrow \mathcal{G}_{\boldsymbol{S}} \geq \mathcal{G}_0.$$

We denote the set of such $\boldsymbol{S}$ by $\mathcal{S}_0$, and $\mathcal{S}_1$ the set of $\boldsymbol{S}$ s.t. $dHSIC(\mathbb{P}_{\varepsilon}\boldsymbol{s}) > 0$.

Denote $0 < M := \min_{S_1 \in \mathcal{S}_1} dHSIC(\mathbb{P}_{\varepsilon}s_1)$. For each $\boldsymbol{S}_1 \in \mathcal{S}_1, \forall \epsilon > 0$, the following must hold for sufficiently large $n$,

$$P\left(\left|\widehat{dHSIC}(\hat{\varepsilon}_1^{\boldsymbol{S}_1}, \ldots, \hat{\varepsilon}_d^{\boldsymbol{S}_1}) - dHSIC(\mathbb{P}_{\varepsilon}s_1)\right| < \frac{M}{2}\right) \geq 1 - \epsilon.$$

Therefore by a union bound, for sufficiently large $n$,

$$P\left(\left|\widehat{dHSIC}(\hat{\varepsilon}_1^{\boldsymbol{S}_1}, \ldots, \hat{\varepsilon}_d^{\boldsymbol{S}_1}) - dHSIC(\mathbb{P}_{\varepsilon}s_1)\right| < \frac{M}{2}, \forall \boldsymbol{S}_1 \in \mathcal{S}_1\right) \geq 1 - C_d \epsilon$$

$$\implies P\left(\widehat{dHSIC}(\hat{\varepsilon}_1^{\boldsymbol{S}_1},\ldots,\hat{\varepsilon}_d^{\boldsymbol{S}_1}) > \frac{M}{2}, \forall \boldsymbol{S}_1 \in \mathcal{S}_1\right) \geq 1 - C_d\epsilon$$

for some constant $C_d$ regarding $d$. Similarly, for a fixed $S_0 \in \mathcal{S}_0$, $\forall \epsilon$, $\exists M_\epsilon$ s.t. for sufficiently large $n$,

$$P\left(\widehat{dHSIC}(\hat{\varepsilon}_1^{\boldsymbol{S}_0},\ldots,\hat{\varepsilon}_d^{\boldsymbol{S}_0}) < M_\epsilon(r(n) + n^{-1/2}) < \frac{M}{2}\right) \geq 1 - \epsilon.$$

Thus, for sufficiently large $n$, the estimated graph $\hat{\mathcal{G}}_n$ induced by the global minimizer of Equation (3) satisfies

$$
\begin{aligned}
&P(\hat{\mathcal{G}}_n \geq \mathcal{G}_0)\\
=&P\left(\exists S_0 \in \mathcal{S}_0 \text{ s.t. } \forall S_1 \in \mathcal{S}_1 , \widehat{dHSIC}(\hat{\varepsilon}_1^{\boldsymbol{S}_0} \ldots,\hat{\varepsilon}_d^{\boldsymbol{S}_0}) < \widehat{dHSIC}(\hat{\varepsilon}_1^{\boldsymbol{S}_1},\ldots,\hat{\varepsilon}_d^{\boldsymbol{S}_1})\right)\\
\geq&P\left(\text{For a fixed } S_0 \in \mathcal{S}_0, \forall S_1 \in \mathcal{S}_1 , \widehat{dHSIC}(\hat{\varepsilon}_1^{\boldsymbol{S}_0} \ldots,\hat{\varepsilon}_d^{\boldsymbol{S}_0}) < \widehat{dHSIC}(\hat{\varepsilon}_1^{\boldsymbol{S}_1},\ldots,\hat{\varepsilon}_d^{\boldsymbol{S}_1})\right)\\
\geq&P\left(\text{For a fixed } S_0 \in \mathcal{S}_0, \forall S_1 \in \mathcal{S}_1 , \widehat{dHSIC}(\hat{\varepsilon}_1^{\boldsymbol{S}_1},\ldots,\hat{\varepsilon}_d^{\boldsymbol{S}_1}) > \frac{M}{2}, \right.\\
&\qquad \left. \text{and } \widehat{dHSIC}(\hat{\varepsilon}_1^{\boldsymbol{S}_0},\ldots,\hat{\varepsilon}_d^{\boldsymbol{S}_0}) < M_\epsilon(r(n) + n^{-1/2})\right)\\
\geq&1 - (1 + C_d)\epsilon,
\end{aligned}
$$

where we also used the union bound in the last inequality.

$\square$

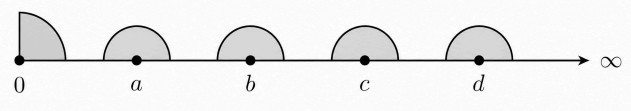

*Figure 4.* Illustration of the 3rd step. Here, $0$ is the theoretical dHSIC statistic induced by $\boldsymbol{S} \in \mathcal{S}_0$, and $a, b, c, d$ are the theoretical dHSIC statistics induced by $\boldsymbol{S} \in \mathcal{S}_1$. The grey regions represent the set of values whose distance to some theoretical dHSIC statistics is less than $M/2$. For sufficiently large $n$, they are separate. By the union bound, with high probability, the estimated statistics are concentrated around the corresponding theoretical statistics i.e. fall into the grey regions. This means that, with high probability, the minimizer would be chosen from $\mathcal{S}_0$.

*Remark* C.1 (Remark on Assumption 3.3). Here we provide more details on the first condition. We first briefly recall the standard empirical-process argument underlying the consistency of empirical risk minimization, c.f. Van Der Vaart & Wellner (1996). Let

$$Z = (Y, X), \qquad \ell_f(Z) = \{Y - f(X)\}^2,$$

and define the empirical and population risks

$$R_n(f) = P_n\ell_f = \frac{1}{n}\sum_{i=1}^{n}\{Y_i - f(X_i)\}^2, \qquad R(f) = P\ell_f = \mathbb{E}\{Y - f(X)\}^2.$$

Let $\hat{\mathcal{F}}$ be the candidate regression class and let

$$f^* \in \arg\min_{f \in \hat{\mathcal{F}}} R(f)$$

denote the population risk minimizer, i.e. the best approximation to the true regression function within $\mathcal{F}$, which is not necessarily $E[Y|X]$. Suppose that the induced loss class

$$\mathcal{L} = \left\{(y, x) \mapsto (y - f(x))^2 : f \in \hat{\mathcal{F}}\right\}$$

is Glivenko–Cantelli. That is,

$$\sup_{f \in \hat{\mathcal{F}}} |R_n(f) - R(f)| = \sup_{f \in \hat{\mathcal{F}}} |P_n\ell_f - P\ell_f| \xrightarrow{p} 0. \tag{8}$$

Assume moreover that the population risk has a well-separated minimizer in the function metric of interest, c.f. Theorem 5.7 of Van Der Vaart (1998). For example in the sup-norm, for every $\varepsilon > 0$,

$$\inf_{\substack{f \in \hat{\mathcal{F}}: \\ \|f - f^*\|_\infty \geq \varepsilon}} \left\{ R(f) - R(f^*) \right\} > 0.$$

Let $\hat{f}_n$ be an empirical risk minimizer, or more generally, an approximate empirical risk minimizer satisfying

$$R_n(\hat{f}_n) \leq \inf_{f \in \hat{\mathcal{F}}} R_n(f) + o_p(1).$$

Then the usual argmin-consistency argument implies

$$\|\hat{f}_n - f^*\|_\infty \xrightarrow{p} 0. \tag{9}$$

In the following, we show that $\mathcal{L}$ is Glivenko–Cantelli when $\hat{\mathcal{F}}$ is some class of neural networks. In fact, let $\hat{\mathcal{F}}$ be the NN class considered in Equation (4) of Schmidt-Hieber (2020) whose covering number was derived in Lemma 5 of that paper. Since $\forall f \in \hat{\mathcal{F}}$ is bounded, $\mathbb{E}Y^2 < \infty$ implies that the squared loss class admits the integrable envelope

$$\ell_f(Y, X) = (Y - f(X))^2 \leq (|Y| + B)^2, \qquad \mathbb{E}(|Y| + B)^2 < \infty.$$

Thus a sup-norm cover of $\hat{\mathcal{F}}$ induces an $L_1$-cover of the squared loss class $\mathcal{L}$. Combining the integrable envelope with the finite covering number implies that $\mathcal{L}$ is Glivenko–Cantelli, c.f. Theorem 2.4.3 of Van Der Vaart & Wellner (1996). Consequently, Equation (8) and Equation (9) hold. This shows that the first condition of Assumption 3.3 is standard in statistical learning.

## D. Experimental details

**Computation.** All experiments were conducted on 12 Intel Xeon Platinum 8360Y processors with 12 cores, 2.40 GHz frequency, and 10GB of memory for each core. For GraN-DAG (Lachapelle et al., 2020), we use 20GB memory for $d = 50$ and 30GB memory for $d = 70$ on a NVIDIA P100 GPU. The wall time is 24 hours.

**Noise distributions.** We consider three noise distributions:

- Standard normal distribution with zero mean and variance 1.

- Centered lognormal distribution with scale parameter 1.

- Standard Student $t$ distribution with degrees of freedom parameter 3.

**Baseline methods.**

- CAM(Bühlmann et al., 2014): Implementation available at Kalainathan et al. (2020). We use the default hyperparameters.

- NOTEARS-MLP (Zheng et al., 2020): Implementation available at `https://github.com/xunzheng/notears`.

- DAGMA (nonlinear) (Bello et al., 2022): Implementation available at `https://github.com/kevinsbello/dagma`.

- GraN-DAG (Lachapelle et al., 2020): Implementation available at `https://github.com/kurowasan/GraN-DAG`. We only used the default hyperparameters due to limited computational resources.

**Hyperparameter tuning.** For NOTEARS-MLP and DAGMA, the default choice for the $L_1$ penalty $\lambda$ is 0.01, and the default choice for the cutoff threshold is 0.3 (i.e. all values less than 0.3 in the estimated adjacency matrix are set to 0). To identify the oracle hyperparameter, we select $\lambda \in \{0.005, 0.01, 0.05, 0.1, 0.2, 0.5, 1, 2, 5\}$, and the cutoff threshold in $\{0.05, 0.1, 0.3, 0.5\}$ that give the best SHD results. All other hyperparameters are set to the default ones.

**NOTIME-CAM configuration.** Here we describe the configuration of NOTIME-CAM in more detail.

- The adaptive group lasso penalty: The default choice is $\lambda = 1.0$. For the oracle hyperparameter, we select $\lambda \in \{0.005, 0.01, 0.05, 0.1, 0.2, 0.5, 1, 2, 5\}$.

- Cutoff threshold: The default choice is the 30% quantile of the adjacency matrix. Since we have additional pretraining steps and use the $\widehat{dHSIC}$ instead of the MSE, the range of the parameters in the learned MLP is different from that of DAGMA and NOTEARS-MLP. For the oracle hyperparameter, we select it in $\{5\%, 10\%, 30\%, 50\%\}$ quantiles.

- $\gamma$: It is discussed in Berrevoets et al. (2025) that the hyperparameter $\gamma$ is necessary to reduce the $O(n^{-1})$ bias of $\widehat{dHSIC}$. Here, we slightly adjust their original proposal to adapt to nonlinear ANMs.

*Table 4.* Choices of $\gamma$ in the experiments.

| $d$ | 10 | 30 | 50 | 70 |
|---|---|---|---|---|
| $\gamma$ | 10 | $1e^4$ | $1e^{16}$ | $1e^{16}$ |

- All other hyperparameters are either the same as NOTEARS or selected as described in Section 5.

# E. Additional results

## E.1. FDR and TPR for default hyperparameters $m_1 = 100$

Here, we provide the false discovery rate (FDR) and the true positive rate (TPR) corresponding to the results in Figures 2 and 3.

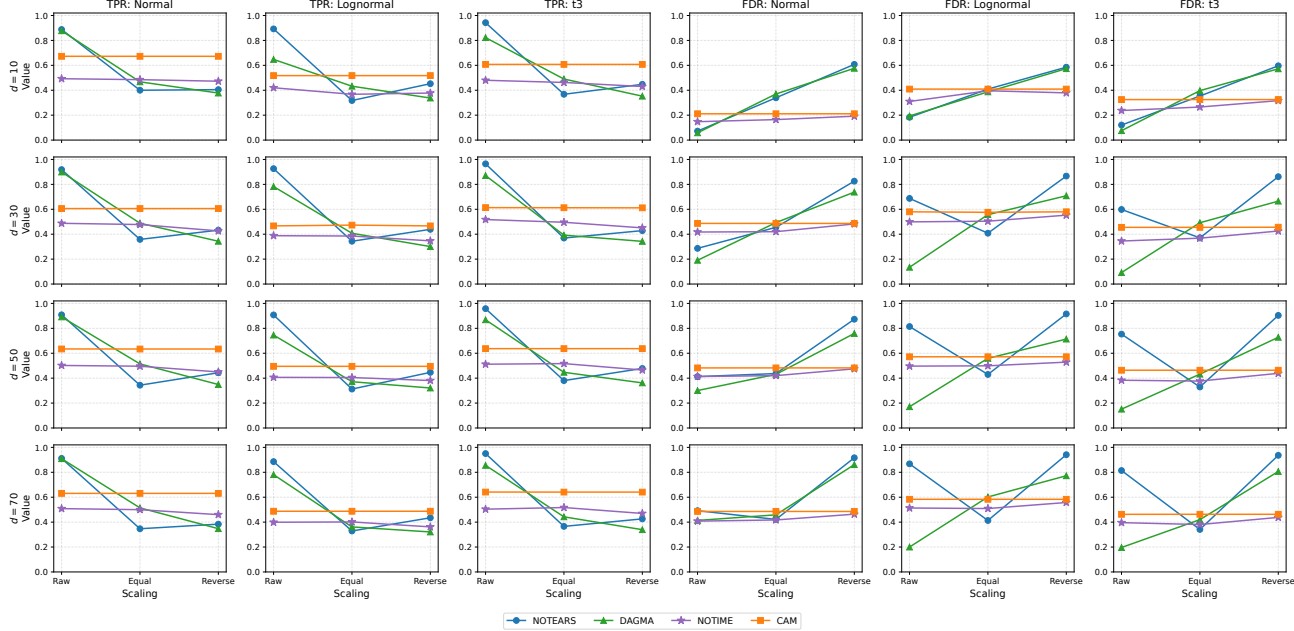

*Figure 5.* TPR (higher better, columns 1-3) and FDR (lower better, columns 4-6) for $d = 10, 30, 50, 70$ with default hyperparameters. From left to right: Results for normal, lognormal, and $t(3)$ noise distribution. X-axis: Scaling scheme. Y-axis: TPR/FDR.

## E.2. SHD for oracle hyperparameters $m_1 = 100$

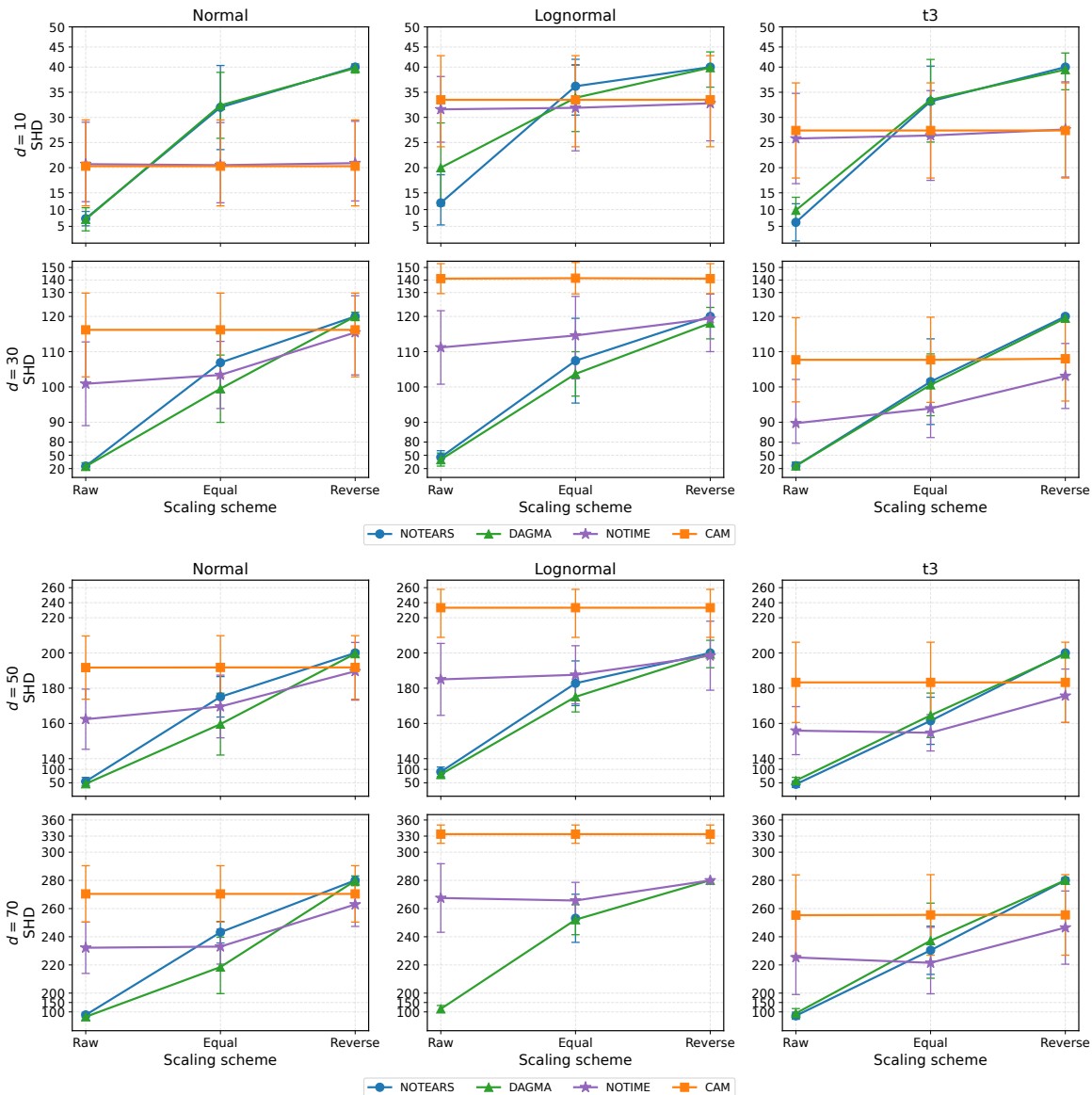

*Figure 6.* Mean SHD (lower better) to the ground truth for $d = 10, 30, 50, 70$ with oracle hyperparameters. From left to right: Results for normal, lognormal, and $t(3)$ noise distribution. X-axis: Scaling scheme. Y-axis: SHD. Error bars based on the standard deviation.

**E.3. SHD for oracle hyperparameters** $m_1 = 10$

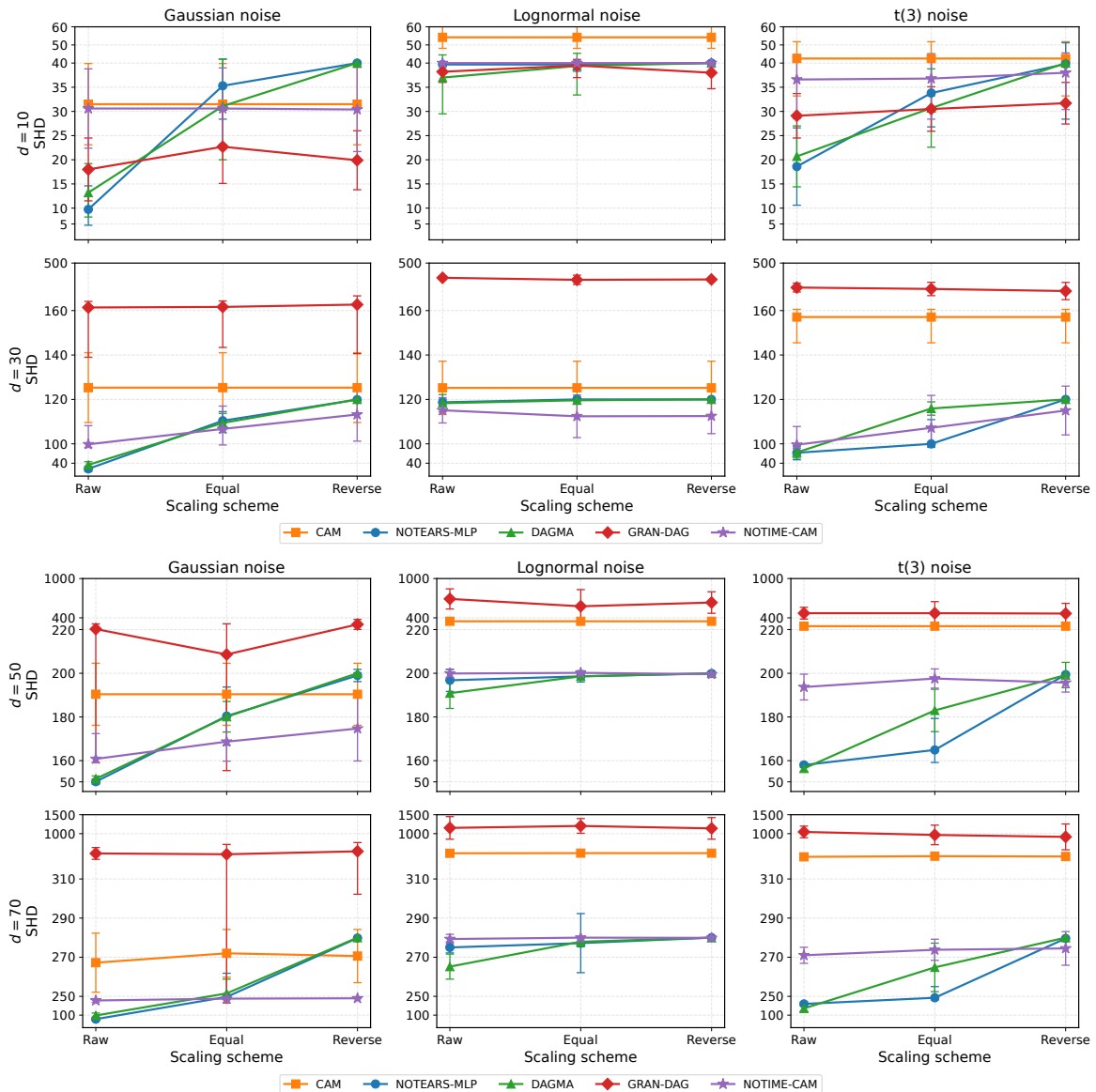

*Figure 7.* Mean SHD (lower better) to the ground truth for $d = 10, 30, 50, 70$ with oracle hyperparameters. From left to right: Results for normal, lognormal, and $t(3)$ noise distribution. X-axis: Scaling scheme. Y-axis: SHD. Error bars based on the standard deviation. We see that NOTIME-CAM is the best method for normal noise and performs well for lognormal noise. For $t(3)$ noise, it is the best method for the reversely scaled data.

### E.4. Ablation studies:

| Noise / Rescale | Variant 1 | | | Variant 2 | | | Variant 3 | | | NOTIME-CAM | | |
|---|---|---|---|---|---|---|---|---|---|---|---|---|
| | Raw | Equal | Reverse | Raw | Equal | Reverse | Raw | Equal | Reverse | Raw | Equal | Reverse |
| Normal | 30.4 | 36.4 | 40.0 | 30.5 | 35.8 | 40.0 | **29.4** | 31.2 | 32.1 | 30.6 | **30.6** | **30.4** |
| Lognormal | **39.6** | 40.0 | 40.0 | 39.8 | 39.9 | 40.0 | 44.2 | 44.6 | 45.6 | 40.0 | 40.0 | 40.0 |
| $t(3)$ | 37.9 | 39.2 | 39.8 | 39.2 | 39.3 | 40.0 | **36.5** | 37.4 | 38.1 | 36.6 | **36.8** | **38.0** |

*Table 5.* Results for $d = 10$ under different rescaling schemes and variants. Data is generated by a two-layer MLP with a hidden dimension $m_1 = 10$. For fair comparisons, results are based on the oracle hyperparameters.

### E.5. Tuning thr_adapt

| $d$ | Quantile, $\eta$ | Raw | Equal | Reverse |
|---|---|---|---|---|
| 10 | 10%, 0.1 (default) | 30.6 | **30.6** | **30.4** |
| | 50%, 0.1 | **30.5** | **30.6** | 31.3 |
| | 10%, 1 | 41.2 | 42.5 | 42.0 |
| 50 | 10%, 0.1 (default) | 160.8 | **168.7** | 174.7 |
| | 50%, 0.1 | 161.8 | 169.0 | **174.0** |
| | 10%, 1 | **160.7** | 169.4 | 174.5 |

*Table 6.* Sensitivity analysis with respect to quantile threshold and $\eta$. Data is generated by a two-layer MLP with a hidden dimension $m_1 = 10$. For fair comparisons, results are based on the oracle hyperparameters.

