# OpenReview forum: "Identifiable Nonlinear Differentiable Causal Discovery via Independence and Adaptive Group Sparsity"
_ICML.cc/2026/Conference — ICML 2026 regular_

### Official Review · Reviewer_s2ua · 2026-03-04

**Soundness:** 2
**Presentation:** 3
**Significance:** 2
**Originality:** 2
**Overall Recommendation:** 4
**Confidence:** 3

**Summary:**

This paper proposes a differentiable approach for causal discovery in nonlinear additive noise models (ANMs), focusing on identifying the true underlying DAG structure from observational data—regardless of noise distribution or variance. The method extends the NOTIME framework to nonlinear settings by minimizing a dependence criterion among residuals, with theoretical analysis showing recovery of the correct DAG up to constant edges. To prune such constant or non-informative edges, the authors introduce an adaptive group lasso penalty on the first-layer neural network weights, leveraging a CAM-based initialization for improved convergence. Empirical results on synthetic data demonstrate that the proposed method (NOTIME-CAM) outperforms or rivals previous differentiable approaches, especially in settings with heterogeneous or non-Gaussian noise.

**Compliance With Llm Reviewing Policy:**

Affirmed.

**Final Justification:**

The authors' reply has resolved part of my concerns, so I think it is reasonable to moderately increase my score.

**Key Questions For Authors:**

1. Can you provide runtime/memory comparisons between the proposed method and baselines, especially for d ≥ 50? How does the increased regularization or initialization complexity affect practical scalability?
1. Have you tested the approach on observational real-world datasets or semi-synthetic data? If not, what challenges do you foresee in such scenarios, especially regarding non-additive noise, latent confounders, or dependent noise structures?

**Limitations:**

yes

**Strengths And Weaknesses:**

**Strengths**
1. Provides a principled framework for nonlinear DAG discovery that is theoretically justified even with arbitrary noise distributions and variances.
1.The adaptive group lasso penalty is well-motivated and, when combined with CAM-based initialization, addresses real challenges in pruning non-informative edges, with practical significance for interpretability and edge support recovery (see Equation 5 and corresponding discussion).
1. Extensive and systematic experiments comparing against strong baselines (CAM, NOTEARS-MLP, DAGMA, GraN-DAG) on synthetic datasets and under varied noise types and normalization (Table 2), showing consistently robust performance.
1. The results tables (Table 2 and Table 3) support claims about both overall accuracy and stability, reporting structural Hamming distances and their variances.

**Weaknesses**
1. While population-level identifiability is rigorously demonstrated, the paper does not address finite-sample rates, required sample sizes, or sensitivity to sample size, noise distribution, or function class misspecification. This is acknowledged as a limitation, but some quantitative or empirical sensitivity analysis would increase practical relevance.
1. All empirical results are on simulated datasets, following standard practice for theory-motivated DAG learning but limiting assessment of practical performance and real-world challenges. A small-scale real-data experiment, even as a “stress test”, would be valuable.
1. There is no quantitative comparison of computational cost (runtime/memory) across methods, which may be consequential for high-dimensional settings.
1. The method shows residual SHD vs. the true graph (e.g., Table 2) even for moderate dimensions/dataset size. While the results outperform most baselines, they do not fully reach the ground truth—further qualitative or case-specific insight into these remaining errors would be useful.

---

> ### Author Rebuttal · Authors · 2026-03-31
>
> **W1. The paper does not address finite-sample rates, required sample sizes, or sensitivity to sample size, noise distribution, or function class misspecification.**
>
> We acknowledge that there are many open questions regarding nonlinear causal discovery and its sensitivities to the underlying data-generating process and potential misspecifications. We would like to highlight that a condition for studying the effect and sensitivity to sample size makes the most sense when there is **a guarantee of identification, which our method unlocks**.
>
> * To accommodate your request, we have complemented a theoretical analysis of finite-sample convergence. Due to space limits, we refer to our reply to **Q3 of Reviewer 2uw7 for the assumptions and proof**.
>
>     ***Theorem:*** Let $\hat\varepsilon_j=X_j-\hat E[X_j|X_{S_j}]$ and $\varepsilon_j=X_j-E[X_j|X_{S_j}]$ for some parent set $S_j$ induced by a causal graph. Under certain assumptions,
>    1. $\widehat{dHSIC}(\hat\varepsilon_1,\dots,\hat\varepsilon_d)\overset{P}{\rightarrow}dHSIC(\varepsilon_1,\dots,\varepsilon_d)$ at rate $O_P(r(n)+1/\sqrt{n})$.
>    2. The global minimum of $\widehat{dHSIC}$ recovers the underlying DAG with the same rate.
>
> * Regarding the sensitivity to the type of noise distribution, the simulation studies consider different distributions for the noise, which included **normal, lognormal and t3 noise and different scaling of the data in Section 4 of the paper**.
>
> * Regarding the sensitivity to model misspecification: It is unfortunately hard to quantify theoretically without precisely characterizing the type of misspecification, which inevitably limits the scope of such an analysis. **Empirically, we have verified that mild misspecifications do not influence the results substantially.** Below we show the SHD of the following experiment. The true underlying function is a 2-layer neural network with $100$ hidden units. The data is of dimension $d=10$, and different choices of noise distribution and scaling method are considered. For fitting, we only used $10$ hidden units. This is a clear misspecification, but we did not see a major difference in the results in general, potentially because the magnitude of the underlying function has been increased by using more neurons, and therefore the signal-to-noise ratio is also affected.
>
>    |Noise|raw|equal|reverse|
>    |:-|-|-|-|
>    |Normal|19.8|20.0|19.7|
>    |Lognormal|31.3|32.3|32.5|
>    |t3|24.8|25.4|27.3|
>
> **Actions taken:** Added the results to Section 3 and the appendix.
>
> **W2/Q2. All empirical results are on simulated datasets...A small-scale real-data experiment would be valuable.**
>
> We have compared NOTIME-CAM with competitors on the causal discovery benchmark "Sachs" [1], with $n=853$ cases, $11$ variables and $17$ edges.
>
> |Method | Parameters | SHD |
> |:-|-|-|
> |CAM|Default|30|
> |**NOTIME-CAM**|Default|**15**|
> |NOTEARS|Default|22|
> |DAGMA|Default|17|
> |**NOTIME-CAM**|Optimal|**14**|
> |NOTEARS|Optimal|16|
> |DAGMA|Optimal|16|
>
> NOTIME-CAM outperforms all competitors under both default parameters ($\lambda_1=1,\lambda_2=0.01,thr\_{cut}=0.1$) and optimal parameters, while CAM performs poorly. This shows that (i) CAM itself can perform poorly and (ii) **even when initialized by a poor CAM fit, NOTIME can still substantially improve it**.
>
> **Actions:** Added the above discussion to Section 4.
>
> **W3/Q1. Computational cost across methods, which may be consequential for high-dimensional settings...especially d >= 50**
>
> Below, we calculate the median runtime $\pm$ IQR/2:
>
> |d|$t_{CAM}$|$t_{init}$|$t_{train}$|
> |:-|-|-|-|
> |10|78.79 $\pm$ 7.86|1.53 $\pm$ 0.33|396.37 $\pm$ 177.21|
> |50|721.22 $\pm$ 38.65|4.18 $\pm$ 0.25|676.03 $\pm$ 576.66|
>
> $t_{train}$ is comparable to CAM for $d=50$. Moreover, NOTIME-CAM only needs $10GB$ in all experiments. Therefore, the computational cost is worth it for the identifiability guarantees.
>
> **Actions:** Added the discussion to the Appendix.
>
> **W4. While the results outperform most baselines, they do not fully reach the ground truth—further qualitative or case-specific insight into these remaining errors would be useful.**
>
> It is indeed true that the ground truth is not always reached. There are several potential explanations for this. First, even the new theoretical finite-sample guarantees only involve high-probability statements which can fail to hold. Second, finding the global minimizer, even in the finite sample case, is not guaranteed. Last, in practice we may have model misspecification which can lead to a persistent bias.
>
> We further analyzed the results on normal noise $d=50$. We see that our method enjoys low FDR and retains the TPR, thus low SHD.
>
> |Method|FDR|||TPR|||
> |:-|-|-|-|-|-|-|
> ||raw|eq|re|raw|eq|re|
> |CAM|0.48|0.48|0.48|0.64|0.64|0.64|
> |**NOTIME-CAM**|0.3|0.33|0.37|0.31|0.29|0.28|
> |NOTEARS|0.07|0.41|0|0.81|0.32|0|
>
> **References:**
> [1] K. Sachs et, al. Causal protein-signaling networks
> derived from multiparameter single-cell data. Science, 2005.

---

> > ### Author Rebuttal · Reviewer_s2ua · 2026-04-04
> >
> > Thank you for your response. I will raise my score.

---

> > > ### Author Response · Authors · 2026-04-04
> > >
> > > Thank you very much for your positive feedback. We are pleased that our rebuttal addressed your concerns, and we greatly appreciate your decision to raise the score. Your thoughtful comments and suggestions have been very helpful in improving the paper.

---

### Official Review · Reviewer_2uw7 · 2026-03-10

**Soundness:** 3
**Presentation:** 3
**Significance:** 2
**Originality:** 3
**Overall Recommendation:** 4
**Confidence:** 3

**Summary:**

This paper addresses the problem of causal discovery in nonlinear additive noise models (ANMs) using a differentiable optimization framework. The authors extend the NOTIME framework, originally designed for linear models, to nonlinear settings by minimizing a multivariate independence measure (dHSIC) of residuals. They provide theoretical guarantees that the global minimizer of this objective recovers the true causal minimal DAG up to constant edges. To address optimization challenges, they propose a CAM-based initialization scheme and an adaptive group lasso penalty to prune non-informative edges. Empirical results on synthetic datasets demonstrate competitive or superior performance compared to existing differentiable methods, especially under heteroscedastic and non-Gaussian noise.

**Compliance With Llm Reviewing Policy:**

Affirmed.

**Final Justification:**

The rebuttal addressed my main concerns, but it did not change my overall evaluation of the paper. Therefore, I maintain my recommendation for weak accept.

The paper has a solid theoretical foundation in terms of soundness, and the proposed method can identify the true causal structure in nonlinear additive noise models. However, it lacks an analysis of finite-sample behavior, which could be a limitation in practical applications. In terms of originality, while extending the NOTIME framework to nonlinear settings is a novel contribution, the idea of using independence-based loss is not entirely new, and the core innovation of the method relies heavily on CAM initialization, which may limit its broader applicability. Regarding significance, the research fills an important gap in nonlinear causal discovery and has high academic value, but the improvements over existing methods are modest, and the reliance on initialization remains a potential issue. As for clarity, the paper is generally well-structured, but certain sections, such as the proof and related work, are somewhat dense. The revisions proposed would help improve this aspect. Overall, while there are some limitations, the research is significant and has elements of originality, making it worthy of further refinement before acceptance.

I recommend that the authors revise the paper to address the concerns raised about statistical testing, finite-sample guarantees, and clarification of their contributions.

**Key Questions For Authors:**

1. In the bivariate example (Section 2.1), how do you ensure that the function class used for regression in the anti-causal direction is not misspecified? Could this explain the lower MSE for the wrong direction?
2. The adaptive group lasso penalty is based on the L2 norm of first-layer columns. How sensitive is the method to the choice of threshold `thr_adapt`? Is there a principled way to tune it?
3. The proof of Theorem 3.1 relies on oracle independence measures and population-level identifiability. Do you have any finite-sample guarantees or concentration results for the dHSIC-based loss?
4. How does the method perform when CAM fails to provide a reasonable initialization (e.g., in high-dimensional sparse settings)? Is the method robust to poor initialization?

**Limitations:**

Yes, the authors discuss limitations such as the additive noise assumption, the lack of finite-sample theory, and the reliance on good initialization. These are appropriately acknowledged.

**Strengths And Weaknesses:**

**Soundness:**

+ The theoretical claim that minimizing an oracle independence measure identifies the true causal ordering is well-grounded in the identifiability literature of ANMs.
+ The proof outline is plausible, though the reliance on oracle measures and population-level arguments leaves finite-sample behavior unaddressed.
+ The empirical setup is thorough, covering multiple noise types, scaling methods, and graph sizes. However, the absence of statistical significance tests (e.g., error bars, confidence intervals) weakens the empirical claims.
+ The adaptive group lasso and CAM-based initialization are sensible design choices, but their impact is not rigorously isolated or ablated.

**Presentation:**

+ The paper is generally well-structured, but some sections are dense and difficult to follow (e.g., the proof sketch and related work discussion).
+ The narrative could be improved by clearly separating theoretical contributions from algorithmic heuristics.
+ The notation is sometimes inconsistent

**Significance:**

+ The problem of identifiable nonlinear causal discovery is important and timely.
+ The proposed method addresses a clear gap in differentiable causal discovery: the lack of identifiability guarantees in nonlinear settings.
+ However, the improvements over baselines are modest in many settings, and the method still relies heavily on CAM for initialization, which may limit its novelty and impact.

**Originality:**

+ The extension of NOTIME to nonlinear ANMs is novel, though the core idea of using independence-based loss is not entirely new (cf. DARING).
+ The adaptive group lasso and CAM-based initialization are pragmatic innovations, but their contribution is incremental.
+ The theoretical insight that constant edges are a unique challenge in nonlinear settings is valuable.

---

> ### Author Rebuttal · Authors · 2026-03-30
>
> We appreciate the reviewer’s insightful feedback. Please find our reply below.
>
> **W1. The absence of statistical significance tests (e.g., error bars, confidence intervals).**
>
> Appendix C reports standard deviations of SHD across repetitions in Table 3. The standard deviation of NOTIME-CAM is comparable to others. When NOTIME-CAM performs better, competing methods are usually beyond or close to the error-bar boundary, supporting the effectiveness of our method.
>
> **Actions:** Merged SHD and error-bar results in Section 4.
>
> **W2/Q4. The adaptive group lasso and CAM-based initialization...their impact is not rigorously isolated or ablated.**
>
> We conducted ablation studies with three variants: (1) $L_1$ penalty + random initialization, (2) group lasso penalty + random initialization, and (3) CAM initialization + $L_1$ penalty. No variant dominates the others in $d=10$, and none outperforms our proposal. Results for $d=50$ will be added and are expected to be qualitatively similar.
>
> |Noise|raw|eql|re|raw|eq|re|raw|eq|re|
> |:-|-|-|-|-|-|-|-|-|-|
> ||variant1|||variant2|||variant3|
> |Normal|**30.4**|36.4|40.|30.5|**35.8**|40.|40.3|40.|40.|
> |Lognormal|39.6|40.|40.|39.8|39.9|40.|**29.2**|**29.3**|**29.**|
> |t3|**37.9**|**39.2**|39.8|**39.2**|39.3|40.|40.0|40.1|40.|
>
> We further see in our reply to **W1/W2 of Reviewer bZfK that even when initialized by a poor fit, NOTIME can still substantially improve it.**
>
> **W3. Some sections are dense and difficult to follow...could be improved.**
>
> **Actions:** We now rephrased the related work and emphasized that they do not provide multivariate identifiability guarantees, motivating our paper. We also separated identifiability guarantees from the algorithm and unified notation.
>
> **Q1. In Section 2.1, how do you ensure...is not misspecified?**
>
> In Example 2.1, we don't require the anti-causal regression function to be correctly specified. Let $\hat f$ be the fitted model. Then by the bias-variance decomposition,
> $$E[(X_1-\hat f(X_2))^2]\ge E[(X_1-E[X_1|X_2])^2].$$
> Thus, with a correctly specified anti-causal model class, the MSE could only be even smaller. Since the true bivariate nonlinear ANM is identifiable, this shows that MSE is inconsistent and motivates methods with identifiability guarantees.
>
> **Actions:** Clarified this point in Example 2.1.
>
> **Q2. How is the adaptive group lasso penalty sensitive to the choice of threshold *thr_adapt*? Is there a principled way to tune it?**
>
> In differentiable causal learning, there is no universally accepted tuning method. It may be based on sample size (NOTEARS) or empirical results in small settings (GraN-DAG). We follow GraN-DAG for this parameter.
>
> We now test *thr_adapt* as the 10% or 50% quantile of $\ell_{ij}$ times a constant $\eta$ under normal noise for different $d$. The default value performs well, suggesting robustness.
>
> |d|quantile,$\eta$|raw|equal|reverse|
> |:-|-|-|-|-|
> |10|10%,0.1(default)|30.6|**30.6**|**30.4**|
> ||50%,0.1|**30.5**|**30.6**|31.3|
> ||10%,1|41.2|42.5|42.|
> |50|10%,0.1(default)|160.8|**168.7**|174.7|
> ||50%,0.1|161.8|169.0|**174.0**|
> ||10%,1|**160.7**|169.4|174.5|
>
> **Q3. Finite-sample guarantees for the dHSIC-based loss**
>
> We now provide finite-sample guarantees for NOTIME below.
>
> ***Assumptions:***
> 1. For $X \in \\{X_1,\dots,X_d\\}$ and $X_S\subset\\{X_1,\dots,X_d\\}\backslash\{X\}$, $\hat E[X|X_S]$ converges uniformly to some $\hat f(X_S)$ at rate $O_P(r(n))$.
> 2. $\hat f$ and the noise class satisfy condition 19 of Peters 2014.
>
> ***Theorem:*** Let $\hat\varepsilon_j=X_j-\hat E[X_j|X_{S_j}]$ and $\varepsilon_j=X_j-E[X_j|X_{S_j}]$ for some parent set $S_j$ induced by a causal graph. Then:
> 1. $\widehat{dHSIC}(\hat\varepsilon_1,\dots,\hat\varepsilon_d)\overset{P}{\rightarrow}dHSIC(\varepsilon_1,\dots,\varepsilon_d)$ at rate $O_P(r(n)+1/\sqrt{n})$.
> 2. The global minimum of $\widehat{dHSIC}$ recovers the underlying DAG with the same rate.
>
> ***Proof:***
>
> 1. $\widehat{dHSIC}(\varepsilon_1,\dots,\varepsilon_d)-dHSIC(\varepsilon_1,\dots,\varepsilon_d)=O_P(1/\sqrt{n})$. Indeed, $\widehat{dHSIC}$ differs from the empirical U-statistic of $dHSIC$ by only $O_P(1/n)$ uniformly by Lemma C.2 of Pfister et al. (2018). The U-statistic then converges to population $dHSIC$ at rate $O_P(1/\sqrt n)$ by Theorem A in Section 5.6 of [1].
>
> 2. Uniform convergence of the regression estimator gives $\|(\hat\varepsilon_1-\varepsilon_1,\dots,\hat\varepsilon_d-\varepsilon_d)\|_2=O_P(r(n))$. Since the kernel is Lipschitz and bounded, so is $\widehat{dHSIC}$, thus
> $$\widehat{dHSIC}(\hat\varepsilon_1,\dots,\hat\varepsilon_d)-dHSIC(\varepsilon_1,\dots,\varepsilon_d)=O_P(r(n)+1/\sqrt{n}).$$
>
> 3. By the 2nd assumption and $dHSIC=0$ iff the residuals are independent, a union bound over all finite topological orders yields $P(\hat G_n\neq G_0)\to0$. $\blacksquare$
>
> **Actions:** Added the finite-sample guarantees to Section 3.
>
> **Reference:**
>
> [1] Serfling, R. J. (1980). *Approximation Theorems of Mathematical Statistics*. John Wiley and Sons.

---

> > ### Author Rebuttal · Reviewer_2uw7 · 2026-04-06
> >
> > The rebuttal has addressed my concerns.

---

> > > ### Author Response · Authors · 2026-04-07
> > >
> > > Thank you very much for your thoughtful comments and for carefully considering our rebuttal. We are very glad that your concerns have been addressed, and we sincerely appreciate your feedback, which has helped us improve the paper.
> > >
> > > If you feel that an updated score would better reflect your current view after the rebuttal, we would sincerely appreciate your consideration.

---

### Official Review · Reviewer_bZfK · 2026-03-12

**Soundness:** 3
**Presentation:** 3
**Significance:** 3
**Originality:** 2
**Overall Recommendation:** 4
**Confidence:** 2

**Summary:**

The paper proposes NOTIME-CAM, a differentiable causal discovery method for nonlinear additive noise models (ANMs) with arbitrary noise distributions and heteroscedasticity. Unlike existing approaches that rely on MSE or likelihood objectives, the method minimizes a residual independence measure (dHSIC) under a DAG constraint and theoretically shows that the global minimizer recovers the true causal DAG up to constant edges. To remove these extra edges, the authors introduce an adaptive group lasso penalty on the first layer neural network weights to enforce edge level sparsity, and use a CAM-based initialization to provide a good causal ordering and stabilize optimization. Experiments on synthetic datasets with various noise types and scaling schemes demonstrate that the proposed approach is robust and competitive with existing differentiable causal discovery methods.

**Compliance With Llm Reviewing Policy:**

Affirmed.

**Final Justification:**

The questions raised in the rebuttal have been mostly addressed and I am content. The experiments on Sachs does show good promise to real applicability. I have increased my score to 4.

**Key Questions For Authors:**

1. The proposed method relies on CAM to obtain the causal ordering and initialize the neural network. How sensitive is the final performance to errors in the CAM ordering or skeleton? For example, if CAM makes significant mistakes in the initial graph, does NOTIME-CAM still recover the correct structure?

2. Theoretical results are shown at the population level. Do the authors have any theoretical or empirical insights into the finite-sample behavior of the method?

3. The experiments consider graphs up to 70 variables. How does the method scale to larger graphs (100–500 nodes) in terms of computational cost and memory, especially given the dHSIC computation and neural network training?

4. Could the authors provide an ablation study showing the individual contribution of (a) the adaptive group lasso penalty and (b) the CAM-based initialization? This would help clarify which component contributes most to the performance improvements.

5. The experiments focus on synthetic data. Do the authors have results that evaluate the method on real world causal discovery benchmarks, and would the method require additional tuning for such settings?

**Limitations:**

Yes.

**Strengths And Weaknesses:**

Strengths

1. The paper clearly identifies a limitation of MSE-based differentiable causal discovery under heteroscedastic or non-Gaussian noise and provides theoretical justification for using an independence-based objective.
2. Theorem showing that minimizing residual dependence recovers the true DAG up to constant edges is a meaningful extension of prior work (NOTIME) to nonlinear ANMs.
3. The adaptive group lasso penalty aligns well with the neural architecture by enforcing sparsity at the edge level rather than parameter level.
4. Experiments demonstrate stable performance across different noise distributions and variable scaling settings, highlighting advantages over several existing differentiable methods.

Weaknesses

1. Experiments are restricted to synthetic datasets. Evaluation on real-world datasets would strengthen the practical relevance.
2. The method relies heavily on CAM initialization, raising questions about robustness when the initial ordering is inaccurate.
3. The overall approach largely combines existing components (NOTIME objective, CAM initialization, group lasso), making the novelty primarily in integration rather than a fundamentally new framework.
4. Identifiability holds at the population level, but there is no analysis of finite-sample behavior or optimization guarantees.

---

> ### Author Rebuttal · Authors · 2026-03-30
>
> We appreciate the reviewer's insightful feedback. Please find our reply below.
>
> **W1/Q5: Evaluation of the method on real world causal discovery benchmarks.**
>
> We have compared NOTIME-CAM with competitors on the causal discovery benchmark "Sachs" [1], with $n=853$ cases, $11$ variables and $17$ edges.
>
> |Method | Parameters | SHD |
> |:-|-|-|
> |CAM|Default|30|
> |**NOTIME-CAM**|Default|**15**|
> |NOTEARS|Default|22|
> |DAGMA|Default|17|
> |**NOTIME-CAM**|Optimal|**14**|
> |NOTEARS|Optimal|16|
> |DAGMA|Optimal|16|
>
> NOTIME-CAM outperforms all competitors under both default parameters ($\lambda_1=1,\lambda_2=0.01,thr\_{cut}=0.1$) and optimal parameters, while CAM performs poorly. This shows that (i) CAM itself can perform poorly and (ii) **even when initialized by a poor CAM fit, NOTIME can still substantially improve it**.
>
> In differentiable causal learning, now there is *no universally accepted tuning method*. It may be based on sample size (NOTEARS) or empirical results in small settings (GraN-DAG); similar heuristics also apply to non-differentiable methods (PC, CAM). We therefore follow this convention and leave in-depth tuning studies for future work.
>
> **Actions:** We have added the above discussion to Section 4.
>
> **W2/Q1: The method relies heavily on CAM initialization.**
>
> As seen in “Sachs”, our method substantially improves the CAM initialization. There, CAM has 8 false (+) and 6 false (-) due to a wrong topological ordering, while NOTIME makes no ordering error for the optimal parameter. This suggests that **NOTIME strongly improves bad initialization**. The experiments (Section 4) and the ablation study in Q4 support this result. We therefore regard it as a **weak-to-moderate start**.
>
> **Actions:** Emphasized this point in the text.
>
> **W3. The overall approach largely combines existing components.**
>
> Our primary goal was to **highlight and solve the fact that common objective functions are fundamentally flawed**. This makes our contribution threefold: (i) differentiable nonlinear causal discovery with identifiability guarantees, (ii) quantitative initialization from a non-differentiable causal learning method, and (iii) incorporation of the adaptive group lasso penalty for the neural network structure.
>
> **Actions:** Emphasized the main novelty of our paper in Section 1.
>
> **W4/Q2: Analysis of finite-sample behavior or optimization guarantees**
>
> We now provide finite-sample guarantees for NOTIME in a new theorem.
>
> ***Theorem:*** Let $\hat\varepsilon_j = X_j - \hat E[X_j|X_{S_j}]$, and $\varepsilon_j = X_j - E[X_j|X_{S_j}]$ for some parent set $S_j$ of $X_j$ induced by a causal graph. Under certain assumptions, we have
> 1. $\widehat{dHSIC}(\hat\varepsilon_1,\dots,\hat\varepsilon_d)\overset{P}{\rightarrow}dHSIC(\varepsilon_1,\dots,\varepsilon_d)$ with the rate $O_P(r(n) + 1/\sqrt{n})$.
> 2. The global minimum of $\widehat{dHSIC}$ recovers the underlying DAG with the same rate.
>
> Due to space limits, we refer to our reply to **Q3 of Reviewer 2uw7 for the assumptions and proof**.
>
> **Q3: How does the method scale to larger graphs in terms of computational cost and memory?**
>
> We did not include extremely large graphs due to a limitation for all dependence measures, c.f. Berrevoets et al. 2025, and note that DARING also restricts $d$ up to $40$. However, dependence measures provide general identifiability guarantees unavailable for losses such as MSE and log-likelihood. While methods based on them may scale to $d=500$, *the loss of identifiability can invalidate the predictions*. Our goal is thus reliable identification in moderate dimensions.
>
> Below, we calculate the median runtime $\pm$ IQR/2:
>
> |d|$t_{CAM}$|$t_{init}$|$t_{train}$|
> |:-|-|-|-|
> |10|78.79 $\pm$ 7.86|1.53 $\pm$ 0.33|396.37 $\pm$ 177.21|
> |50|721.22 $\pm$ 38.65|4.18 $\pm$ 0.25|676.03 $\pm$ 576.66|
>
> $t_{train}$ is comparable to CAM for $d=50$. Moreover, NOTIME-CAM only needs $10GB$ in all experiments. Therefore, the computational cost is worth it for the identifiability guarantees.
>
> **Actions:** Added the discussion to the Appendix.
>
> **Q4: An ablation study showing the individual contribution of the adaptive group lasso penalty and the CAM-based initialization.**
>
> We conducted ablation studies with three variants: (1) $L_1$ penalty + random initialization, (2) group lasso penalty + random initialization, and (3) CAM initialization + $L_1$ penalty. No variant dominates the others in $d=10$, and none outperforms our proposal. Results for $d=50$ will be added and are expected to be qualitatively similar.
>
> $d=10$
> |Noise|raw|eql|re|raw|eq|re|raw|eq|re|
> |:-|-|-|-|-|-|-|-|-|-|
> ||variant1|||variant2|||variant3|
> |Normal|**30.4**|36.4|40.|30.5|**35.8**|40.|40.3|40.|40.|
> |Lognormal|39.6|40.|40.|39.8|39.9|40.|**29.2**|**29.3**|**29.**|
> |t3|**37.9**|**39.2**|39.8|**39.2**|39.3|40.|40.0|40.1|40.|
>
> **Actions:** Added to the Appendix.
>
> **References:**
> [1] K. Sachs et, al. Causal protein-signaling networks derived from multiparameter single-cell data. Science, 2005.

---

> > ### Author Rebuttal · Reviewer_bZfK · 2026-04-03
> >
> > Thank you for the rebuttal. The questions raised have been mostly addressed and I am content. The experiments on Sachs does show good promise to real applicability. I will increase my score accordingly.

---

> > > ### Author Response · Authors · 2026-04-03
> > >
> > > Thank you very much for your positive follow-up and for taking the time to read our rebuttal carefully. We are glad that our responses addressed your concerns and that you found the Sachs experiments promising for real-world applicability.
> > >
> > > We also appreciate your note that you would increase your score accordingly. Since the review currently still appears unchanged on OpenReview, we just wanted to kindly check in case an update is still pending.
> > >
> > > Thank you again for your time and consideration.

---

### Decision · Program_Chairs · 2026-04-30

**Decision:**

Accept (regular)

**Comment:**

The reviewers aknowledge the novelty of the theoretical contribution, despite some incremental aspects. The main concerns of the reviewers on various aspects were addressed by authors. I therefore recommend acceptance of the paper.